# Nonproductive exposure of PBMCs to SARS-CoV-2 induces cell-intrinsic innate immune responses

Julia Kazmierski[1,2,†] (iD), Kirstin Friedmann[1,†], Dylan Postmus[1,2] (iD), Jackson Emanuel[1] (iD), Cornelius Fischer[3] (iD), Jenny Jansen[1,2], Anja Richter[1], Laure Bosquillon de Jarcy[1,4], Christiane Schüler[1,2], Madlen Sohn[3], Sascha Sauer[3], Christian Drosten[1,5], Antoine-Emmanuel Saliba[6], Leif Erik Sander[4], Marcel A Müller[1,5] (iD), Daniela Niemeyer[1,5,*] (iD) & Christine Goffinet[1,2,**] (iD)

## Abstract

Cell-intrinsic responses mounted in PBMCs during mild and severe COVID-19 differ quantitatively and qualitatively. Whether they are triggered by signals emitted by productively infected cells of the respiratory tract or result from physical interaction with virus particles remains unclear. Here, we analyzed susceptibility and expression profiles of PBMCs from healthy donors upon *ex vivo* exposure to SARS-CoV and SARS-CoV-2. In line with the absence of detectable ACE2 receptor expression, human PBMCs were refractory to productive infection. RT–PCR experiments and single-cell RNA sequencing revealed JAK/STAT-dependent induction of interferon-stimulated genes (ISGs) but not proinflammatory cytokines. This SARS-CoV-2-specific response was most pronounced in monocytes. SARS-CoV-2-RNA-positive monocytes displayed a lower ISG signature as compared to bystander cells of the identical culture. This suggests a preferential invasion of cells with a low ISG baseline profile or delivery of a SARS-CoV-2-specific sensing antagonist upon efficient particle internalization. Together, nonproductive physical interaction of PBMCs with SARS-CoV-2- and, to a much lesser extent, SARS-CoV particles stimulate JAK/STAT-dependent, monocyte-accentuated innate immune responses that resemble those detected *in vivo* in patients with mild COVID-19.

**Keywords** interferon; interferon-stimulated genes; PBMCs; SARS-CoV-2
**Subject Categories** Immunology; Microbiology, Virology & Host Pathogen Interaction
**Mol Syst Biol. (2022) 18: e10961**

## Introduction

The current SARS-CoV-2 pandemic represents a global medical, societal, and economical emergency of increasing importance. Arising at the end of 2019 in the Hubei province in China, the causative agent of the coronavirus disease 2019 (COVID-19), SARS-CoV-2, has to date infected more than 555 million individuals worldwide (World Health Organization). Owing to SARS-CoV-2 infection, more than 6.3 million deaths were reported to date (as of 2022, July 5[th]). The predominant symptoms of symptomatic COVID-19 are fever, cough, and shortness of breath; however, in severe cases, the disease can progress to pneumonia, acute respiratory distress syndrome, and multiple organ failure (Chen *et al*, 2020; Wölfel *et al*, 2020). The management of the pandemic is complicated by a large interindividual spectrum of clinical courses ranging from asymptomatic to fatal outcomes, pre- and asymptomatic infectious phases (Rothe *et al*, 2020; Jones *et al*, 2021), and the ongoing emergence of variants with increased transmissibility and/or immune escape. The reasons for the high interindividual outcome of infection are insufficiently understood and may include different degrees of cross-reactive background immunity at the level of humoral (Anderson *et al*, 2021; Ng *et al*, 2020) and T-cell-mediated immunity (Bacher *et al*, 2020; Braun *et al*, 2020; Nelde *et al*, 2021; Schulien *et al*, 2021), polymorphisms in genes related to innate immunity (Zhang *et al*, 2020) and autoimmunity (Bastard *et al*, 2020). Currently, specific treatment regimens must be administered early postinfection. They include the RNA polymerase inhibitor Remdesivir that may reduce hospitalization time but not mortality (Wang *et al*, 2020b), the protease inhibitor Paxlovid (Hammond *et al*, 2022), the nucleoside analog Molnupiravir and monoclonal anti-spike antibodies with variant-specific neutralization potencies (Weinreich *et al*, 2021; RECOVERY Collaborative

1  Institute of Virology, Campus Charité Mitte, Charité – Universitätsmedizin Berlin, Berlin, Germany
2  Berlin Institute of Health, Berlin, Germany
3  Scientific Genomics Platforms, Laboratory of Functional Genomics, Nutrigenomics and Systems Biology, Max Delbrück Center for Molecular Medicine, Berlin, Germany
4  Department of Infectious Diseases and Respiratory Medicine, Charité - Universitätsmedizin Berlin, Corporate Member of Freie Universität Berlin, Humboldt-Universität zu Berlin, and Berlin Institute of Health (BIH), Berlin, Germany
5  German Center for Infection Research, Associated Partner Charité, Berlin, Germany
6  Helmholtz Institute for RNA-based Infection Research (HIRI), Helmholtz-Center for Infection Research (HZI), Würzburg, Germany
   *Corresponding author. Tel: +0049 30 450 525 488; E-mail: daniela.niemeyer@charite.de
   **Corresponding author. Tel: +0049 30 450 525 489; E-mail: christine.goffinet@charite.de
   †These authors contributed equally to this work as first authors

Group, 2022). In the late phase of infection, the administration of the immune modulator dexamethasone (RECOVERY Collaborative Group, 2021) dampens the hyperactivation of cytokine-driven immune responses. While several effective vaccines are available, the necessity for specific treatment options will likely persist given the expected proportion of the population that will not have access to vaccines or will refuse vaccination.

To accelerate the establishment of immunomodulatory strategies, it is crucial to characterize *ex vivo* systems that correlate with cellular immunophenotypes of SARS-CoV-2 infection *in vivo* and that may contribute to preclinical testing. Furthermore, the usage of *ex vivo* platforms allows the systematic and comparative investigation of human cellular responses to exposure with different representatives of the species SARS-related coronaviruses (CoVs), including SARS-CoV. Peripheral immune cells are major contributors to human cellular responses upon infection. Given the recruitment of blood mononuclear cells to the lung compartment (Bost *et al*, 2020; Delorey *et al*, 2021; Wendisch *et al*, 2021), and the reported presence of viral RNA detectable in the peripheral blood of up to 10% of severely ill patients (Andersson *et al*, 2020; Prebensen *et al*, 2021), direct contact of PBMCs with infectious SARS-CoV-2 virions or defective viral particles is a likely scenario.

Here, we analyzed susceptibility to infection and cell-intrinsic innate responses of peripheral blood cells from healthy donors upon *ex vivo* exposure to SARS-CoV and SARS-CoV-2. Although both SARS-related CoVs failed to detectably replicate and spread in PBMCs, SARS-CoV-2 specifically triggered a JAK/STAT-dependent innate immune response that was most pronounced in monocytes. Single-cell, virus-inclusive RNA sequencing revealed relatively inefficient and ACE2-independent uptake of virus particles and a SARS-CoV-2 exposure-specific gene expression profile. Cellular responses, consisting of upregulation of the expression of interferon-stimulated genes (ISGs) but not proinflammatory cytokines, partially recapitulate expression profiles obtained by single-cell RNA sequencing of PBMCs from patients experiencing mild COVID-19 (Arunachalam *et al*, 2020; Schulte-Schrepping *et al*, 2020; Silvin *et al*, 2020). Our data demonstrate that cells from the peripheral blood, when undergoing contact with SARS-CoV-2 particles, mount cellular responses that potentially contribute to the control and/or pathogenesis of the infection.

# Results

## Absence of productive infection of human PBMCs by SARS-CoV and SARS-CoV-2

To address the ability of SARS-related CoVs to infect and propagate in cells of the peripheral blood, we exposed unstimulated PBMCs from healthy individuals to purified stocks of SARS-CoV and SARS-CoV-2, respectively, using equal infectious titers as determined on Vero E6 cells. As a reference, PBMCs were exposed to supernatants from uninfected Vero E6 cells (mock-exposed). For both SARS-related CoV, infectivity in cell-culture supernatants drastically decreased over time compared with the inoculum, reaching undetectable levels at 3 days postinoculation (Fig 1A), and pointing towards the absence of *de novo* production of infectious particles. Treatment of cells with the polymerase inhibitor Remdesivir did not

further reduce infectivity in the supernatant, suggesting that the infectivity detectable in the mock-treated, virus-exposed cultures reflects virus input (Fig 1B). By contrast, infection of Vero E6 cells with the identical SARS-CoV-2 stock gave rise to a productive and Remdesivir-sensitive infection (Appendix Fig S1). In our experiments, virus-containing supernatant was replaced with a fresh medium 4 h postinoculation. Nevertheless, low levels of viral RNA genome equivalents remained detectable in the culture supernatant until the end of the experiment for both SARS-CoV and SARS-CoV-2 (up to 192 h postexposure; Fig 1C). Viral RNA was abundant also in supernatants from Remdesivir-treated cultures and cultures exposed to heat-inactivated SARS-CoV-2 until 192 h postexposure, arguing for high stability of the residual viral RNA of the inoculum, and against a constant replenishment of extracellular viral RNA pools as a reason for the stable RNA quantities (Fig 1D), in line with reported longevity of the incoming genomic viral RNA (Lee *et al*, 2022). Notably, blunting signaling by type I interferons (IFNs) through the constant presence of the JAK/STAT inhibitor Ruxolitinib failed to enable secretion of infectious particles and viral RNA in the supernatant, suggesting that JAK/STAT-dependent cell-intrinsic innate immunity is not the underlying reason for the absence of detectable virus production (Fig 1A and C).

To elucidate if PBMCs, despite being nonpermissive, are nevertheless susceptible to SARS-related CoV entry and initial RNA replication, we monitored cell-associated viral RNA species in the cultures over time. Because adherence of cells was incomplete before 48 h, we were able to separate adherent and the suspension cell fractions only starting at 72 h postculture start. Cell-associated viral genome equivalents (Fig 1E) and subgenomic viral E and N RNA (Fig EV1), the latter produced during discontinuous viral transcription, remained stable over time, and did not differ quantitatively for both SARS-related CoVs. Ruxolitinib treatment did not detectably enable RNA replication (Figs 1E and EV1), suggesting the absence of essential cofactors at the level of entry and/or RNA replication rather than the antiviral activity of IFN-regulated restriction factors. In line with this idea, we failed to detect the expression of the SARS-CoV receptor, angiotensin-converting enzyme 2 (ACE2) in PBMCs, as judged by immunoblotting, flow cytometry, and Q-RT–PCR using ACE2-specific antibodies and primer/probes, respectively (Fig EV2A–C). Also, we failed to detect relevant quantities of NRP-1 expression by flow cytometry (Fig EV2D), which has been suggested as an alternative entry receptor in conditions of low-to-absent ACE2 abundance (Cantuti-Castelvetri *et al*, 2020; Daly *et al*, 2020). In conclusion, freshly isolated, unstimulated PBMCs are devoid of expression of ACE2 and putative alternative receptor NRP-1. Furthermore, they appear to be nonsusceptible and nonpermissive to infection with either SARS-related CoV, at least *ex vivo*. However, the continuous presence of viral RNA associated with cells and in the culture supernatant suggests that virus particles attach to and/or internalize into PBMCs in an ACE2-independent manner and remain cell-associated for up to several days.

## Exposure of PBMCs to SARS-CoV-2 and, to a much lower extent SARS-CoV, triggers a JAK/STAT-dependent cell-intrinsic innate immune response

To identify potential cell-intrinsic innate immune responses to SARS-CoV and SARS-CoV-2 exposure, we analyzed *IFIT1* and *IL6*

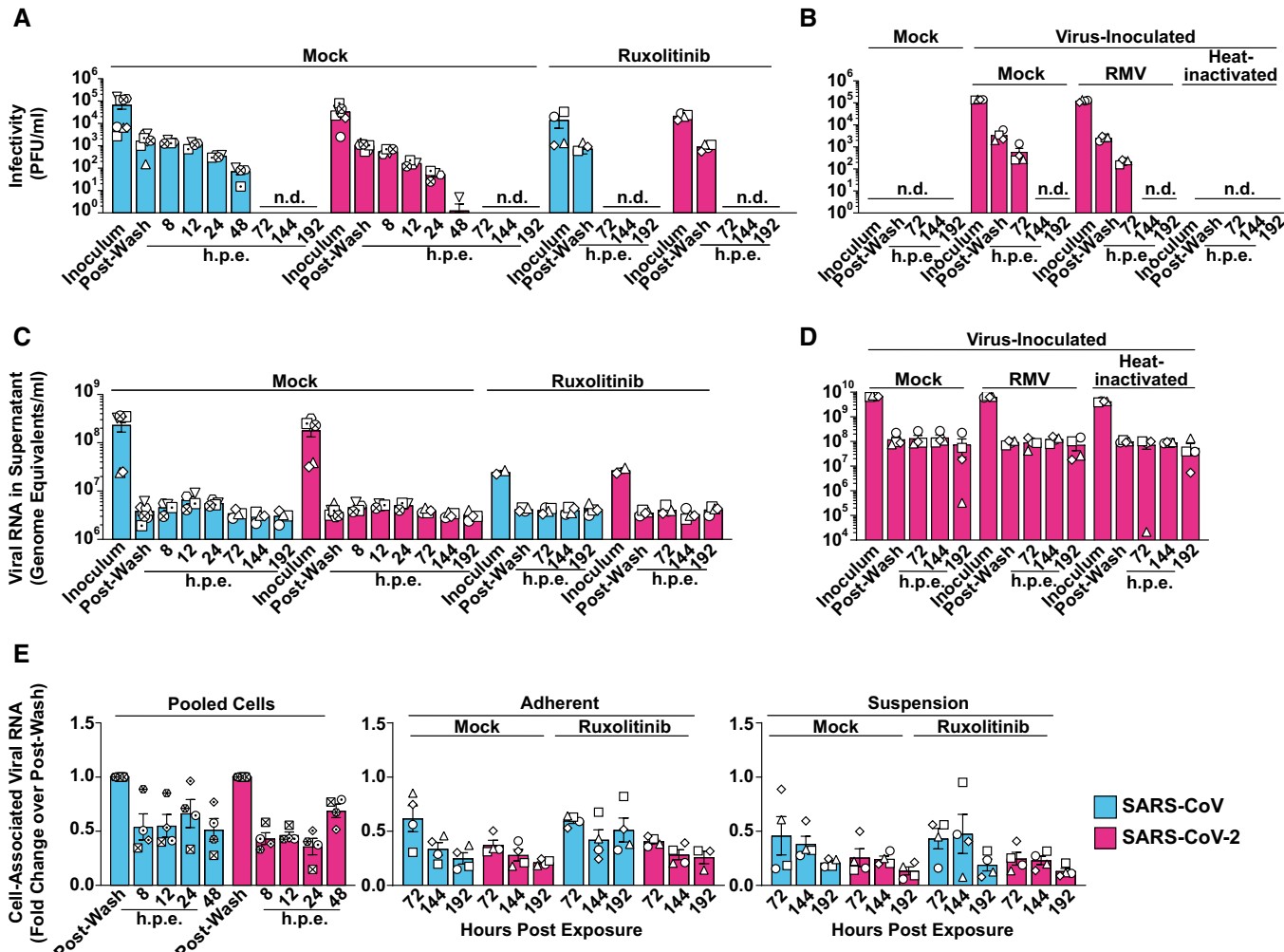

**Figure 1. Absence of productive infection of PBMCs by SARS-CoV and SARS-CoV-2.**

Untreated or Ruxolitinib (10 μM)-treated PBMCs from four individual donors were exposed to SARS-CoV or SARS-CoV-2 (MOI 0.5). PBMCs inoculated with supernatant from Vero E6 cell cultures mixed with PBS and OptiPro serum-free medium supplemented with 0.5% gelatine were used as a control condition (Mock). Supernatants and individual cell fractions were collected at indicated time points postinoculation and analyzed for:

A, B    Infectivity in cell-culture supernatants by plaque titration assay.

C, D    Viral RNA (genome equivalents/ml) concentrations in cell-culture supernatants by Q-RT–PCR.

E    Relative changes of cell-associated viral genomic RNA quantities by Q-RT–PCR and normalized to *RNASEP* levels.

Data information: Data were generated in four individual experiments using cells from at least four individual donors represented by different symbols, bars represent the mean, and error bars indicate the SEM. Statistical significance was tested using the paired Student's *t*-test comparing mock- and Ruxolitinib-treated samples. *P*-values > 0.05 were considered not significant and are not shown in the figure. n.d., not detectable; h.p.e., hours postexposure; RMV, Remdesivir; Ruxo., Ruxolitinib.

mRNA expression over time (Figs 2A and EV3). We selected *IFIT1* and *IL6* as prototypic target genes that are transcribed by IRF3- and NF-κB-dependent promoter activation, respectively (Honda & Taniguchi, 2006). In contrast to SARS-CoV-inoculated cells, SARS-CoV-2-exposed cells displayed Ruxolitinib-sensitive, significantly upregulated *IFIT1* mRNA expression at 16, 24, and 48 h postinoculation (Fig 2A). Inhibition of potential low-level SARS-CoV-2 RNA replication through treatment of cells with Remdesivir, and heat inactivation of the SARS-CoV-2 stock inoculum did not prevent induction of *IFIT1* mRNA expression (Appendix Fig S2), corroborating the idea that the latter is triggered by exposure to virions but not by productive infection. By contrast, *IL6* expression was barely

induced after exposure to SARS-CoV and SARS-CoV-2 (Fig EV3). Together, SARS-CoV-2 exposure specifically triggered IRF3-induced *IFIT1* but not NF-κB-induced *IL6* gene expression. We next analyzed if type I IFN expression preceded *IFIT1* mRNA expression in SARS-CoV-2-exposed PBMCs. Despite a slight trend for elevated *IFNA1* and *IFNB1* mRNA expression at 16 h, levels failed to reach significant upregulation at 4, 16, and 24 h, when compared to mock-exposed cultures (Fig 2B). However, IFN-α2 and IFN-stimulated IP-10, MCP-1, and MCP-3 proteins, as opposed to IL-6 and several other cytokines (Appendix Fig S3) were secreted in the supernatant of exposed PBMCs in a Ruxolitinib-sensitive manner, with overall higher levels in SARS-CoV-2- than in SARS-CoV-exposed cultures

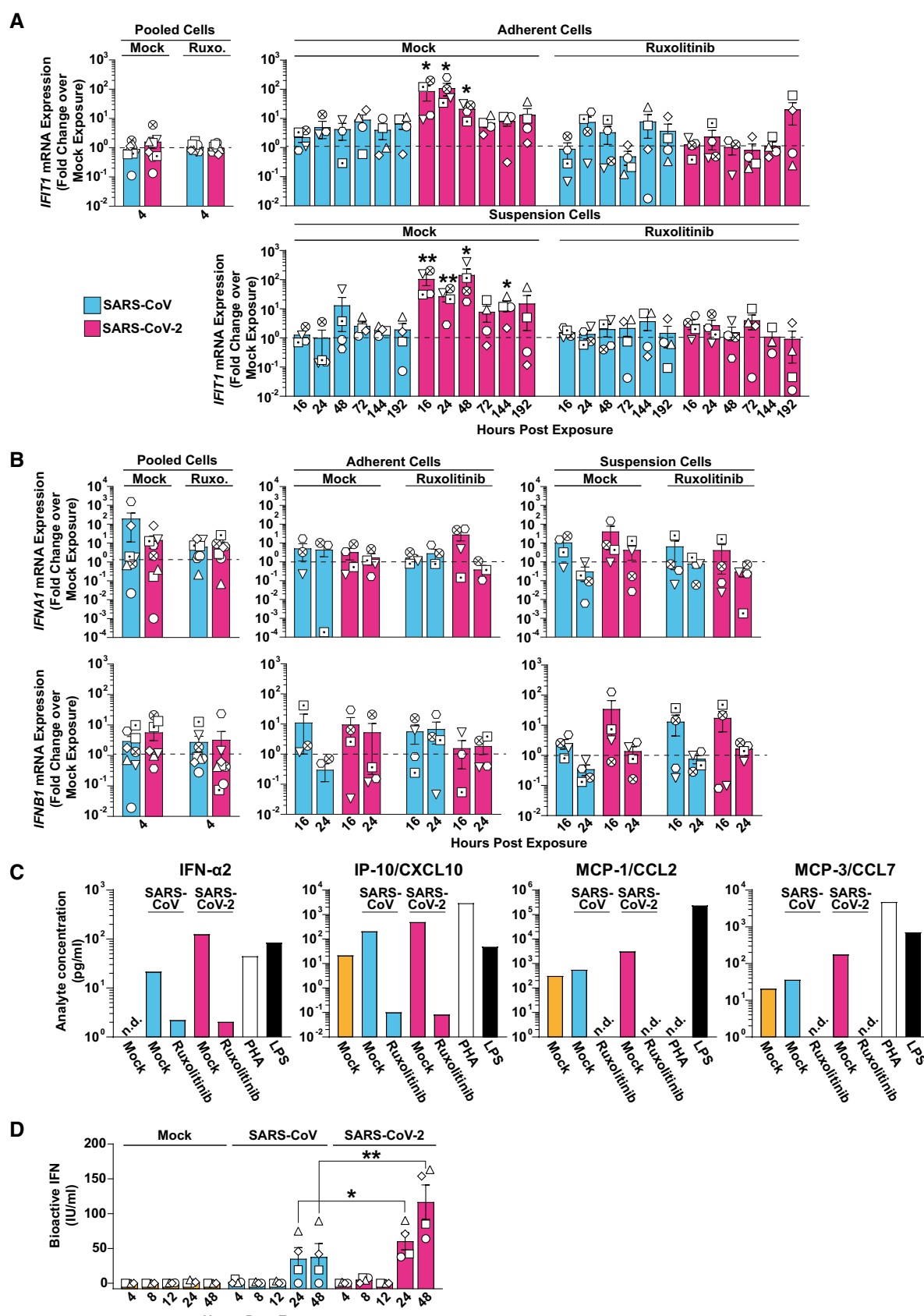

**Figure 2.**

**Figure 2.   Exposure of PBMCs to SARS-CoV-2 and, to a much lower extent SARS-CoV, triggers a JAK/STAT-dependent cell-intrinsic innate immune response.**

A–C     RNA extracted from Ruxolitinib-treated or mock-treated, and SARS-CoV-, SARS-CoV-2-, or mock-exposed PBMCs was analyzed for (A) *IFIT1*, (B) *IFNA1* and *IFNB1*, (C) mRNA expression by Q-RT–PCR at indicated time points. Suspension and adherent cell fractions were analyzed separately, except at the 4 h time point. Values were normalized to cellular *RNASEP* expression and are shown as fold change over mock-inoculated conditions. The dotted line indicates the expression level of mock-inoculated cell cultures and is set to 1. (C) Supernatants from Ruxolitinib- or mock-treated and SARS-CoV-, SARS-CoV-2-, or mock-inoculated PBMCs were collected 48 h postexposure, and cytokine expression of IFN-α2, IP-10/CXCL10, MCP-1/CCL2, and MCP-3/CCL7 were quantified using a Luminex-based immunoassay. PHA- or LPS-treated PBMCs were used as a positive control. Bars represent the results of a pool of four individual samples per condition.

D        Supernatants collected from SARS-CoV-, SARS-CoV-2-, or mock-exposed PBMCs at indicated time points were analyzed for the release of bioactive IFN using the HL116 reporter cell assay.

Data information: Data were generated in four individual experiments using PBMCs from four or more individual donors represented by different symbols, bars represent the mean, and error bars indicate the SEM (A, B, and D). Statistical significance between mock- and Ruxolitinib-treated samples was tested using the paired Student's *t*-test and comparing SARS-CoV with SARS-CoV-2-treated samples from the same donor and time points. *P*-values < 0.05 were considered significant (*), < 0.01 very significant (**), or ≥ 0.05 not significant (not shown). n.d., not detectable.

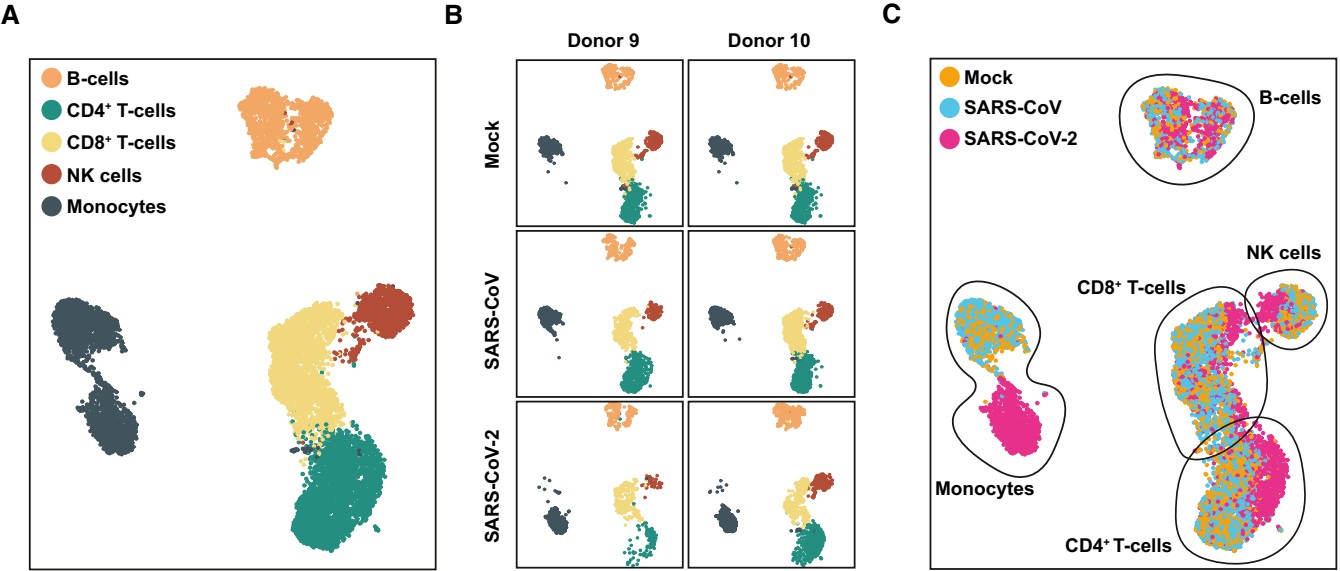

**Figure 3.   SARS-CoV-2 exposure causes transcriptional changes in most cell types.**

PBMCs isolated from two donors were exposed to SARS-CoV, SARS-CoV-2, or mock-exposed, and analyzed by scRNA-sequencing 24 h postexposure.

A     UMAP displaying all identified cell types,

B     UMAP indicating the data obtained from the PBMCs of the two donors,

C     Identified cell types according to condition.

(Fig 2C). Furthermore, bioactive IFN was detected in the supernatant of corresponding cultures, with highest levels upon SARS-CoV-2 exposure, while SARS-CoV inoculation induced the release of bioactive IFN at inferior levels that were statistically indistinguishable from those in mock-exposed cultures (Fig 2D). These results suggest that, although both SARS-related CoV failed to establish a productive infection in PBMCs, SARS-CoV-2 appears to induce cell-intrinsic, IFN-mediated, and JAK/STAT-dependent responses in several cell types comprised in PBMCs. By contrast, SARS-CoV induced very mild, if any, innate immune responses.

## SARS-CoV-2 exposure causes transcriptional changes in most cell types

To explore cell-intrinsic responses in individual cell types, we performed single-cell RNA sequencing of PBMCs exposed to SARS-CoV and SARS-CoV-2, respectively. We identified the five major cell types, namely B cells, CD4[+] and CD8[+] T cells, NK cells, and monocytes (Fig 3A) based on the expression of discriminatory marker mRNAs (see Materials and Methods). Separated based on experimental conditions, PBMCs of both donors shared a similar relative cell type distribution (Fig 3B) and similar cell type-specific transcriptional profile (Appendix Fig S4), and data of both donors were merged for the following analyses. In line with our bulk analyses (Fig EV2A–C), *ACE2* mRNA was undetectable, as was *TMPRSS2* mRNA (Fig EV2E). By contrast, the protease-encoding *FURIN*, *BSG*, and *NRP1* mRNAs were expressed in all cell types and most abundantly in monocytes (Fig EV2E). Graphical mapping indicated transcriptomic changes within individual cell types for SARS-CoV-2-exposed, but not for SARS-CoV-exposed cultures, compared with mock-inoculated cells (Fig 3C). Notably, SARS-CoV-2 monocytes clustered separately from the other conditions in the UMAP despite library batch correction, implying a pronouncedly altered transcriptome. The T- and NK-cell clusters slightly and partially shifted, indicating a change in their

transcriptional profile (Fig 3C). The relative abundance of T cells and monocytes in SARS-CoV-2-exposed cells as compared to mock-exposed PBMCs remained constant, as judged by flow cytometric analysis (Appendix Fig S5). Together, this analysis revealed that transcriptomic changes occurred in most cell types upon SARS-CoV-2 exposure, particularly in the monocytic fraction.

## Exposure to SARS-CoV-2 induces a global innate immunity-related gene profile in PBMCs with cell type-specific signatures

We next investigated in more detail the cell type-specific response to SARS-CoV-2. For each cell type, cells from all three treatments were subclustered and genes differentially expressed between clusters were used as input for cell trajectory analysis using the Pseudotime algorithm from the monocle R package (Trapnell *et al*, 2014). We aimed to identify whether cells from different treatments, especially those exposed to different viruses, developed along the same trajectory as a result of the exposure or if a different cell fate was induced (Fig 4A). For all five major cell types, cells inoculated with SARS-CoV-2 developed towards a separate cell fate and largely branched off from mock-exposed and SARS-CoV-exposed cells, which, conversely, shared a common trajectory. Interestingly, B-cell analysis resulted in four branching points, from which only two (#1 and #3) were specific for SARS-CoV-2-exposed cells, suggesting a high transcriptional heterogeneity of B cells independently of virus exposure. Though progression through pseudotime resulted in a distinct and highly pronounced trajectory of all SARS-CoV-2-exposed cell types, this effect was most clear in monocytes (Fig 4A). Analysis of expression of specific genes, including *ISG15* and *IFIT1*, confirmed that in general, all cell types contributed to gene expression changes upon SARS-CoV-2 challenge, and monocytes displayed the most pronounced elevation of expression of both genes (Fig 4B). Identification of differentially expressed genes (DEGs) in mock-exposed compared with SARS-CoV-2-inoculated PBMCs revealed a significant upregulation of gene expression in all cell types, especially in monocytes (Fig 4C). Interestingly, the majority of DEGs were identified as known ISGs (defined by the interferome database; colored in green (interferome.org; v2.01)). Scoring the individual cell types and conditions by their expression of an IFN-signaling module revealed a SARS-CoV-2-specific induction of ISGs in all cell types, though this was most prominent in monocytes (Fig 4D). Moreover, IFN module scores were colinear with Pseudotime scores along the SARS-CoV-2 trajectory, supporting the notion that SARS-CoV-2 exposure induces a development of PBMCs towards an antiviral phenotype. Increased expression of several ISGs, including

*ISG15*, *IFIT1*, *IFITM3*, *DDX58*, *IFIH1*, *LY6E*, *MX2*, *IFI6*, *BST2*, was detectable predominantly, but not exclusively, in monocytes (Fig 4E), supporting the hypothesis that monocytes play a key role in the induction of cell-intrinsic innate immune response to SARS-CoV-2 stimulation. In line with our previous findings (Fig 2), SARS-CoV-2- and SARS-CoV-exposed cells scored virtually negative for the expression of various cytokines, including *IL6* (Fig 4E) and *IFN* mRNAs (Appendix Fig S6), although they express IFN receptors (Appendix Fig S6). In conclusion, these data reveal a strong induction of cell-intrinsic innate immunity in SARS-CoV-2-exposed PBMCs that manifests predominantly in monocytes.

## Transcriptome differences in viral RNA-positive and bystander monocytes

Next, we aimed at identifying viral RNA-positive cells and their specific transcriptional profile that we hypothesized to differ from cells without detectable viral RNA of the identical culture. SARS-CoV-2 RNA was detectable in all cell types but predominantly in monocytes (Fig 5A). Identified viral reads were distributed over the viral genome sequence, with a high over-representation of the 3' RNA sequences that all subgenomic and genomic viral RNA have in common, corresponding to the 3' part of the N-coding sequence and polyA tail (Fig 5B). Specifically, in SARS-CoV- and SARS-CoV-2-exposed PBMC cultures, we identified 99 (2.13%) and 212 (2.88%) viral RNA-positive cells, respectively (Fig 5C). Among those, we identified 56 (7.8%) and 173 (15.3%) viral RNA-positive monocytes among all monocytes, respectively. First of all, no statistically significant differences in expression of individual genes of RNA-positive and RNA-negative monocytes were identified. However, the IFN module score (Fig 4) was slightly, but statistically highly significantly, elevated in SARS-CoV-2-exposed monocytes with undetectable viral RNA (Fig 5D and E). Specifically, within the 94 genes that were expressed marginally more abundantly in cells lacking detectable SARS-CoV-2 RNA, 18 represented ISGs, including *ISG15*, *IFITM2*, *IFITM3*, *IFI27*, and *HLA* genes that tended to be upregulated in viral RNA-negative bystander cells. Importantly, the presence of viral RNA did not specifically associate with the expression of *BSG/CD147* and *NRP1*, and *ACE2* and *TMPRSS2* expression was undetectable, suggesting that particles internalize in a manner that is independent of these proposed and confirmed receptors, respectively. In SARS-CoV-2 RNA-positive cells as compared to SARS-CoV-2 RNA-negative cells of the identical cultures, among others, *CD163* reads tended to be slightly more abundant. Expression of the hemoglobin-haptoglobin scavenger receptor CD163 has been associated with the regulation of inflammation (Kowal *et al*, 2011) and

**Figure 4.  Exposure to SARS-CoV-2 induces a global innate immunity-related gene profile in PBMCs with cell type-specific signatures.**

A  Pseudotime cell trajectory analysis and GSEA analysis using genes differentially regulated between mock-, SARS-CoV-, and SARS-CoV-2-challenged conditions for indicated cell types.

B  Representative UMAPs showing *IFIT1* and *ISG15* mRNA expression in the indicated conditions.

C  Volcano plot of all DEGs in SARS-CoV-2-exposed cells compared with mock-exposed cells in the indicated cell types. Known ISGs were colored in green based on their presence in the interferome database (http://www.interferome.org/; v2.01).

D  Cell trajectory maps of indicated cell types with cells colored by expression of the genes in an IFN module gene set.

E  Dot plot depicting expression of selected ISGs and cytokines. Expression levels are color-coded, and the percentage of cells expressing the respective gene is coded by symbol size.

Data information: Data shown in this figure are based on the analysis of two donors.

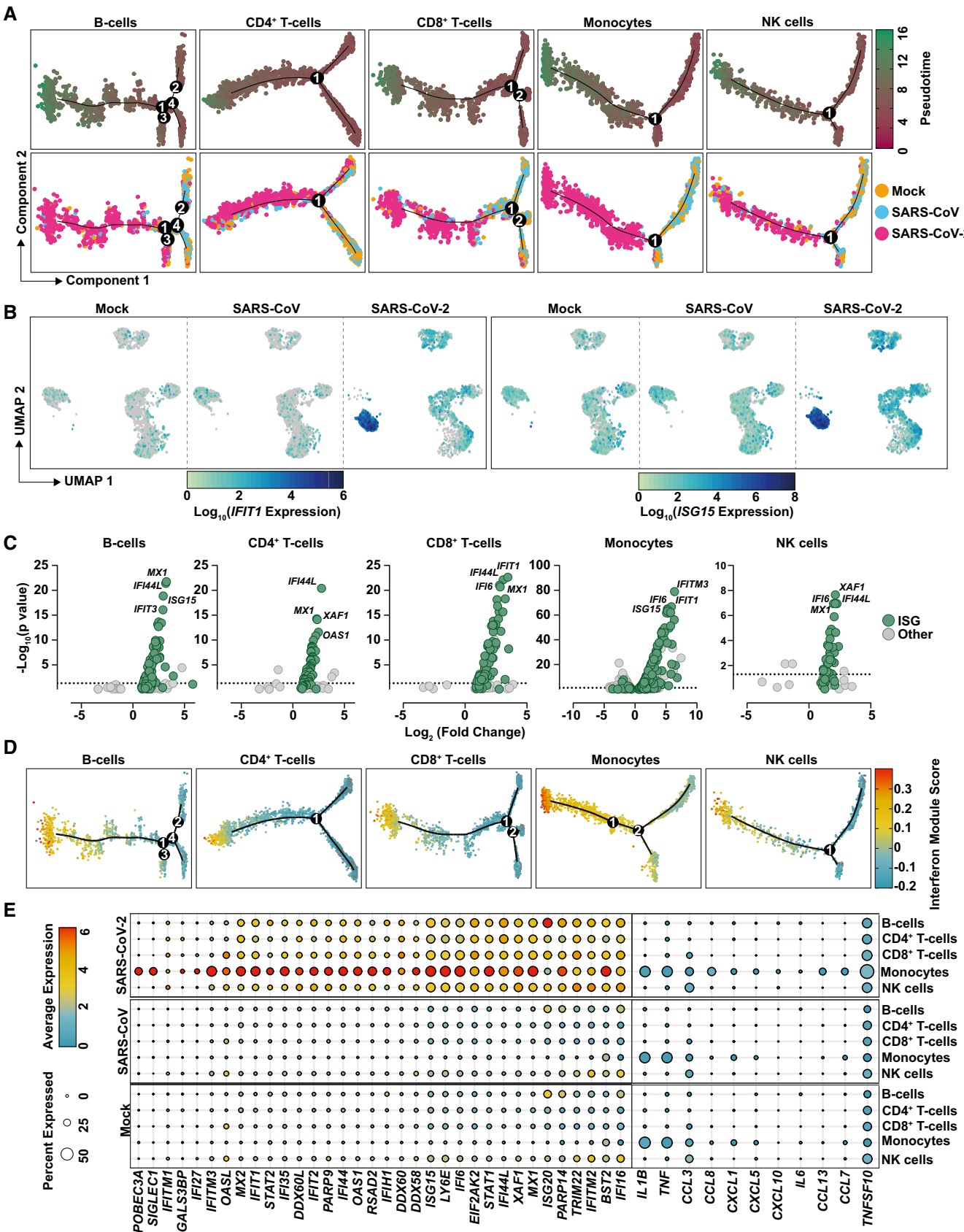

**Figure 4.**

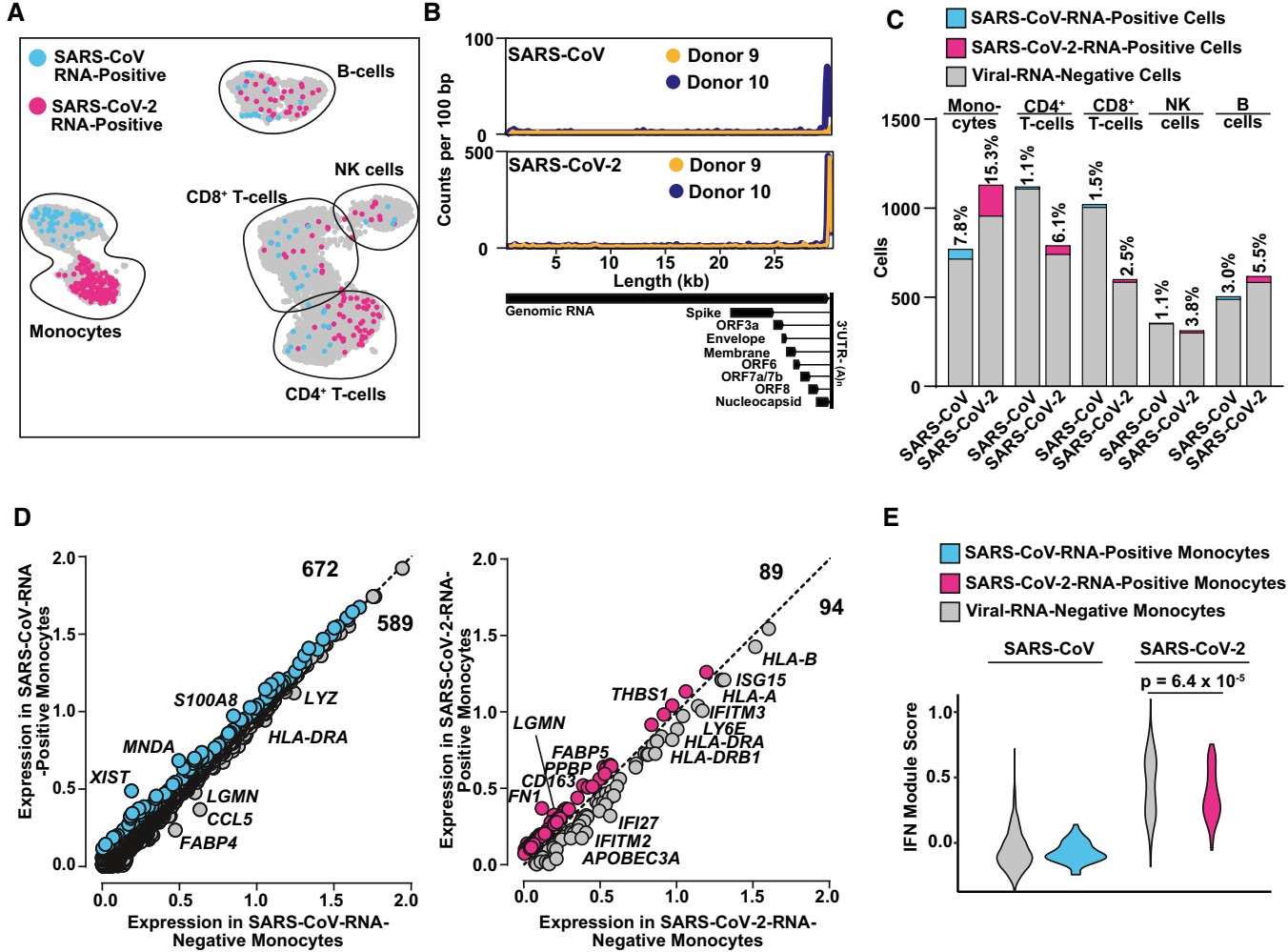

**Figure 5. Viral RNA-positive monocytes trend towards downregulation of ISGs and upregulation of fibrosis-associated genes.**

A  UMAP highlighting cells in which transcripts of either SARS-CoV RNA (blue) or SARS-CoV-2 RNA (magenta) were identified.

B  Virus-specific reads were aligned to the SARS-CoV or SARS-CoV-2 genome. Coverage of the genome is shown in counts per 100 bp.

C  Bar graph showing the absolute number (bars) and relative percentage of cells that were identified as virus RNA-positive in SARS-CoV- and SARS-CoV-2-inoculated cultures, respectively.

D  Plot of Log$_{10}$ average expression of genes showing a Log$_2$(fold change) > 0.2 in viral RNA-positive versus viral RNA-negative monocytes from the SARS-CoV- (left panel) and SARS-CoV-2- (right panel) inoculated PBMCs with genes showing the highest expression fold change between both conditions.

E  IFN Module Score of viral RNA-negative (gray; 715 and 958 cells in SARS-CoV and SARS-CoV-2 exposed cultures, respectively), SARS-CoV-RNA-positive (blue; 56 cells), and SARS-CoV-2-RNA-positive (magenta; 173 cells) monocytes. Statistical significance was tested using a Wilcoxon rank-sum test with continuity correction. *P*-values < 0.05 were considered significant (*), < 0.01 very significant (**), or ≥ 0.05 not significant.

Data information: Data shown in this figure are based on the analysis of two donors.

has interestingly been linked to immunological changes in monocytes and monocyte-derived macrophages from SARS-CoV-2-infected individuals (Gómez-Rial *et al*, 2020; Trombetta *et al*, 2021; Wendisch *et al*, 2021). Looking specifically at the *CD163*^HIGH monocyte population, we found that it displayed high expression levels of genes with profibrotic functions, including *VCAN, LGMN, MERTK, TFGB1, MRC1, TGFBI,* and *MMP9,* and enhanced the expression of cytokines including *CCL2, CXCL8,* or *IL1B* and the cytokine receptor *CCR5* (Fig EV4). Furthermore, SARS-CoV-2 RNA-positive cells displayed a preferential upregulation of genes implicated in migration and integrin binding (*FN1, PPBP, THBS1*), and differentiation, including *FABP5* and *LGMN*. Together, cells that internalized SARS-

CoV-2 particles exhibit a slightly distinct gene expression profile characterized by a consistent reduction in antiviral ISGs and an upregulation of profibrotic genes as opposed to bystander cells with undetectable viral RNA.

Finally, we were intrigued whether preactivation would result in an altered ability of PBMCs to interact with, internalize, and sense SARS-CoV-2, as opposed to freshly isolated PBMCs from healthy donors. To mimic the environment of circulating PBMCs of a SARS-CoV-2-infected individual, we individually pretreated PBMCs with type I IFN, with supernatant from SARS-CoV-2-infected lung epithelial cell cultures, and with serum obtained at an early stage postinfection from mildly COVID-19 diseased individuals (Fig EV5). IFN-

α2a pretreatment did not influence the amount of detectable viral RNA, neither in the cell-associated fraction (Fig EV5A, left panel) nor in the supernatant (Fig EV5B, left panel), despite expected IFN-mediated enhancement of *IFIT1* (Fig EV5C) but not *IL6* mRNA (Fig EV5D) expression.

Pretreatment of PBMCs with virus-free, cytokine-containing supernatant derived from SARS-CoV-2-infected Calu-3 cell cultures resulted in a mild (1.6-fold), but statistically significant increase in cell-associated viral RNA copies (Fig EV5B, middle panel) when compared to cells stimulated with supernatant from uninfected Calu-3 cells, in the absence of changes concerning viral RNA quantities (Fig EV5A, middle panel) in the culture supernatant, and *IFIT1* and *IL6* mRNA expression (Fig EV5C and D, middle panels).

Similarly, cultivation of PBMCs in the presence of serum from COVID-19 patients, as opposed to mock or control serum treatment prior to SARS-CoV-2 exposure, was followed by a 1.8-fold higher abundance of viral genomic RNA in the cellular fraction (Fig EV5B, right panel). In summary, these results suggest that PBMCs that are "primed" by stimulation with a cytokine milieu that is characteristic of an ongoing systemic SARS-CoV-2 infection display a slightly increased ability to physically interact with viral particles, in the absence of detectable changes of *IFIT1* and *IL6* mRNA expression.

## Discussion

In this study we characterized the response of peripheral immune cells, at the cell type level and at individual cells' level, to *ex vivo* SARS-CoV-2 exposure as compared to SARS-CoV. While *ex vivo* experiments inherently do not recapitulate systemic immune cell interactions and lack the context of complex tissues' and organs' interplay and communication, they uniquely allow the side-by-side comparison of two genetically closely related but functionally different viruses under standardized conditions. Furthermore, they allow assessing the direct consequence of virus exposure on individual cell types.

Our results indicate that SARS-CoV-2 and SARS-CoV share the inability to detectably infect PBMCs. Previous studies with SARS-CoV and MERS-CoV yielded partially conflicting results regarding the susceptibility of human PBMCs to infection. *Ex vivo*, one publication reported the absence of SARS-CoV replication in PBMCs (Castilletti *et al*, 2005), while another work suggested susceptibility and permissiveness of PBMCs to SARS-CoV infection with a high interdonor variability (Ng *et al*, 2004). Another report suggested inefficient *de novo* SARS-CoV production in CD14-purified monocytes, despite detectable virus particle uptake, presumably through phagocytosis (Yilla *et al*, 2005). *In vivo*, *in situ* hybridization and electron microscopy analyses reported the presence of SARS-CoV material in lymphocytes and monocytes derived from infected patients (Gu *et al*, 2005). MERS-CoV was suggested to efficiently replicate in *ex vivo*-infected monocytes (Chu *et al*, 2014) but to only abortively infect human T cells (Chu *et al*, 2016). Of note, the confirmed receptor for SARS-CoV-2 cell entry (Hoffmann *et al*, 2020) has been reported to be virtually absent in PBMCs (Song *et al*, 2020; Xiong *et al*, 2020; Zou *et al*, 2020; Xu *et al*, 2020b), a finding that is in line with our own inability to detect *ACE2* mRNA and ACE2 protein expression in PBMCs by various methods. Therefore, we

hypothesize that virus particles attach and/or internalize in an ACE2-independent manner, resulting in viral RNA associated with and/or internalized into cells. Interestingly, prestimulation of PBMCs in a cytokine-containing milieu using virus-free supernatants from infected lung epithelial cell cultures and sera from COVID-19 patients sensitized cells for a slightly more efficient uptake of particles. Given that receptor-independent phagocytosis is a hallmark of monocytes, our observation that the majority of the viral reads were retrieved in monocytic cells underlines this idea. Furthermore, as SARS-CoV ORF7a is a virion-associated protein (Huang *et al*, 2006) and SARS-CoV-2 ORF7a was reported to efficiently interact with PBMC-derived monocytes (Zhou *et al*, 2021), ORF7a may contribute to the attachment to monocytes. Interestingly, the binding capability of SARS-CoV ORF7a protein was reported to be significantly weaker as compared to SARS-CoV-2 ORF7a (Zhou *et al*, 2021), which is consistent with the observed two-fold reduced proportion of virus RNA-positive monocytes in SARS-CoV-exposed PBMCs as compared to SARS-CoV-2.

*In vivo*, a multitude of cytokines, including IL-1β, IL-1RA, IL-7, IL-8, IL-9, IL-10, CXCL10, IFN-γ, and TNF-α are upregulated in the plasma of COVID-19 patients, especially in cases with the severe outcome (Huang *et al*, 2020). By contrast, mild COVID-19 associates with effective type I IFN responses, including expression of type I IFNs themselves and IFN-stimulated genes, which are probably essential to clear the virus infection and orchestrate adaptive immunity accordingly (Arunachalam *et al*, 2020; Schulte-Schrepping *et al*, 2020; Stephenson *et al*, 2021). To date, it remains largely unclear which cell populations are the drivers of these individual responses. Productively infected epithelial cells in the respiratory tract may initiate some of these responses directly; alternatively, or in addition, immune cells may be stimulated by signals released by productively infected cells or by virions and/or viral proteins directly. Studies on the consequence of the physical interaction of SARS-CoV-2 with infection-refractory primary immune cells, as opposed to susceptible cell types in the respiratory tract, are largely missing. Of note, cytokines levels and composition differ in serum and bronchoalveolar lavage fluid of patients with COVID-19 (Xiong *et al*, 2020), suggesting that productively infected epithelial tissue in the respiratory tract and nonsusceptible peripheral immune cells initiate different cytokine responses. Proinflammatory monocytes that infiltrate the lung have been proposed to represent major cytokine producers in the lung microenvironment (Liao *et al*, 2020). In line with this idea, SARS-CoV-2-susceptible infected cell lines and primary cells (Blanco-Melo *et al*, 2020) display imbalanced host responses, with strong cytokine and ablated ISG responses, when compared to other respiratory virus infections. Also, studies performed in the SARS-CoV-2 Syrian hamster model uncovered an early and strong cytokine response in the myeloid compartment of the lung (Nouailles *et al*, 2021). Here, our data provide the first insights into the response of refractory PBMCs upon exposure to virus particles in the absence of co-stimulating infected cell types. The lack of expression of proinflammatory cytokines, including IL-6, TNFa, and IL-1 in SARS-CoV-2-exposed PBMCs, is in line with the idea that these cytokines are mainly derived from the respiratory tract representing the site of productive infection, and it may partially explain the absence of lymphocyte depletion in our experimental setting that is observed *in vivo* (Huang *et al*, 2020; Qin *et al*, 2020; Wang *et al*, 2020a). In our *ex vivo* PBMC setting, which

is devoid of productive infection, SARS-CoV-2 and, to a much lower extent, if at all, SARS-CoV particles induced innate immune responses in the absence of coculture with infected epithelial cells, indicating that direct exposure to virions can trigger responses in PBMCs.

Immune responses were initiated in different cell types with a focus on monocytes and were characterized by ample induction of expression of *IFIT1* and several other ISGs, as opposed to proinflammatory cytokines, including *IL6* mRNA expression. Our data suggest that this response may be triggered, at least to a certain extent, in a virus replication-independent manner. Despite our failure to detect *IFNA1* mRNA expression at the time points investigated, which might be related to its transient presence, the Ruxolitinib-sensitive induction of *IFIT1* mRNA expression, the secretion of IFN-α and IFN-induced cytokines, and bioactive IFN strongly suggest an underlying IFN-signaling-dependent mechanism. This observation is well in line with *ex vivo* data from PBMCs derived from COVID-19 patients showing highest, monocyte-specific induction of IFN-mediated responses that are inversely proportional to the degree of disease (Arunachalam *et al*, 2020; Schulte-Schrepping *et al*, 2020; Stephenson *et al*, 2021), and the absence of proinflammatory cytokine expression (Arunachalam *et al*, 2020; Xu *et al*, 2020a; Stephenson *et al*, 2021). Because our cellular model lacks the complexity of interactions occurring *in vivo* between circulating immune cells and tissue-resident cells, including those of the productively infected respiratory tract, we hypothesize that it approaches the situation in mildly infected individuals with a transient, rapidly controlled phase of virus replication involving a limited amount of virus-induced cell damage and immune dysregulation.

Comparison of cells with and without detectable SARS-CoV-2 RNA revealed quantitative differences regarding gene expression. Genes associated with fibrosis, migration, and integrin binding were mildly upregulated in cells with detectable viral RNA when compared to bystander cells, defined as cells of the SARS-CoV-2-exposed cell culture, which lacked detectable viral reads. Interestingly, monocytes developing profibrotic functions have recently been established in the context of COVID-19 *in vivo* and to be marked by high expression of CD163 (Wendisch *et al*, 2021). The literature suggests that CD163 expression in monocytes and macrophages is tightly regulated by pro- and anti-inflammatory cytokines (reviewed in Etzerodt & Moestrup, 2013) but is also inducible following Toll-like receptor (TLR) signaling, indicating multiple mechanisms of CD163 regulation (Weaver *et al*, 2007). Our *ex vivo* exposure approach does not mirror the cytokine-containing environment monocytes and macrophages of SARS-CoV-2-infected individuals are exposed to, which might, among other reasons, explain the overall mild induction of CD163 in this cellular compartment observed in our data as opposed to data from COVID-19 patients (Gómez-Rial *et al*, 2020; Wendisch *et al*, 2021). Our single-cell RNA-sequencing dataset, however, allows us to speculate about another potential layer of SARS-CoV-2-induced CD163 upregulation in monocytes in the absence of proinflammatory cytokines as a consequence of virus uptake and innate sensing of viral compartments through pattern recognition receptors, such as TLRs.

Bystander cells displayed enhanced ISG expression, suggesting that sensing of viral PAMPs in cells, which internalized virion particles (identified by detectable viral reads) is largely dampened by the delivery of virion-packaged antagonists, whereas cells internalizing virions at a level below our detection limit remain sensing-competent and alert bystander cells, resulting in an elevated IFN module score in the latter. A potentially additional phenomenon is that virion-packaged antagonists lower the overall base-line IFN/ISG level in invaded cells, conversely resulting in a comparably elevated IFN module score in bystander cells. We favor this model because in our experimental set-up, we identified changes of the IFN module score at the single-cell level as a consequence of virus exposure. Interestingly, and in analogy to our findings, uptake of SARS-CoV by CD14-purified monocytes was found to correlate with a low-to-absent baseline level of IFN expression (Yilla *et al*, 2005). However, virus preparations were not cleared from contaminating cytokines originating from the producer cell during virus stock production, and whether the low IFN state was a cause or consequence of SARS-CoV-2 exposure was not investigated in that particular study (Yilla *et al*, 2005), making it difficult to draw analogies to our findings. Indeed, our data cannot exclude the potentially additionally contributing reciprocal scenario of a more efficient and more probable internalization specifically into cells with a low ISG profile, which, however, would imply the existence of an essential, IFN-sensitive step in the uptake of virions that we deem unlikely given the receptor-independent uptake and the nonreproductive nature of the particle uptake. Indeed, IFN treatment prior to SARS-CoV-2 exposure failed to change quantities of viral RNA uptake upon IFN stimulation as judged in a bulk approach. Multiple SARS-related CoV-encoded IFN antagonists, including structural components of the incoming virion that do not require productive infection for expression and function, dampen innate immune responses when ectopically expressed, including membrane and nucleocapsid proteins (Lei *et al*, 2020). In addition, virion components including ORF3- and ORF6-encoded proteins (Bai *et al*, 2021; Cheng Huang *et al*, 2007; Ito *et al*, 2005) have type I IFN evasion properties (Lei *et al*, 2020; Li *et al*, 2020; Schroeder *et al*, 2021). Interestingly, among those, ORF6 from SARS-CoV-2 was described to be inferior in counteracting phospho-IRF3 nuclear translocation in infected cells, compared with SARS-CoV ORF6, resulting in higher ISG induction (Schroeder *et al*, 2021). Therefore, incoming viral RNA sensing may be less efficiently prevented by SARS-CoV-2 ORF6 as compared to SARS-CoV ORF6. Finally, the large absence of a detectable ISG expression profile in SARS-CoV-exposed PBMCs is consistent with a previous report analyzing abortively infected monocyte-derived macrophages (Cheung *et al*, 2005).

By contrast, endemic human CoVs, including 229E, have been shown to actively enter and replicate in blood-derived monocytic cells and macrophages (Desforges *et al*, 2007; Funk *et al*, 2012), in line with the detectable expression of the cellular 229E-specific receptor CD13/APN (Yeager *et al*, 1992; Funk *et al*, 2012) and triggering a strong infection-induced type I interferon responses in the monocytic cell compartments (Cheung *et al*, 2005; Desforges *et al*, 2007). In contrast to 229E, *ex vivo* exposure of monocytes or macrophages to SARS-CoV-2 triggers a type I IFN-dependent response in the absence of productive infection (this manuscript; Zheng *et al*, 2020; Zankharia *et al*, 2022); however, *in vivo* studies clearly demonstrate the contribution of monocytes and macrophages to the SARS-CoV-2-induced disease progression as a consequence of the constant exposure to cytokines and viral PAMPs, eventually resulting in a gradually increasing dysregulated myeloid

cell compartment (Schulte-Schrepping *et al*, 2020; Kosyreva *et al*, 2021; Leon *et al*, 2022). By contrast, the high IFN induction of the low pathogenic HCoV 229E early upon infection, is thought to be beneficial for a rapid, immune-mediated viral clearance, whereas the highly pathogenic HCoVs SARS-CoV and SARS-CoV-2 encode numerous viral antagonists to evade innate signaling, eventually resulting in blunted activation of the host cellular immunity and delayed viral clearance *in vivo* (Fung & Liu, 2019; Kim & Shin, 2021).

Together, our study provides an analysis of gene expression in PBMCs exposed *ex vivo* to SARS-CoV and SARS-CoV-2 at the cell type and individual cell level. Our data suggest that direct stimulation of monocytes through physical contact with SARS-CoV-2 particles is followed by strong ISG induction, despite the absence of detectable productive infection.

# Materials and Methods

## Cell lines and primary cells

Vero E6 (ATCC CRL-1586) cells, Calu-3 (ATCC HTB-55) cells, and HEK293T (ATCC CRL-3216) cells were cultivated in Dulbecco's modified Eagle's medium (DMEM) supplemented with 10% heat-inactivated fetal calf serum, 1% nonessential amino acids (Thermo Fisher Scientific), and 1% sodium pyruvate (Thermo Fisher Scientific) in a 5% $CO_2$ atmosphere at 37°C. Cell lines were routinely monitored for the absence of mycoplasma and paramyxovirus simian virus 5.

Withdrawal of blood samples from healthy humans and cell isolation was conducted with approval of the local ethics committee (Ethical review committee of Charité Berlin, votes EA4/166/19 and EA4/167/19). Human PBMCs were isolated from buffy coats by Ficoll–Hypaque centrifugation. PBMCs were cultured at $2 \times 10^6$/ml in RPMI 1640 containing 10% heat-inactivated fetal calf serum (Sigma Aldrich), 1% penicillin–streptomycin (Thermo Fisher Scientific), and 2 mM L-glutamine (Thermo Fisher Scientific). The experiments conformed to the principles set out in the WMA Declaration of Helsinki and the Department of Health and Human Services Belmont Report.

## Viruses

SARS-CoV isolate HKU-39849 (accession no. JQ316196.1, Zeng *et al*, 2003; van den Worm *et al*, 2012) and the SARS-CoV-2 BetaCoV/Munich/ChVir984/2020 isolate (B.1 lineage, EPI_ISL_406862, Wölfel *et al*, 2020) were used.

Virus was grown on Vero E6 cells and concentrated using Vivaspin® 20 concentrators with a size exclusion of 100 kDa (Sartorius Stedim Biotech) in order to remove cytokines of lower molecular weight, including IFNs. Virus stocks were stored at −80°C, diluted in OptiPro serum-free medium supplemented with 0.5% gelatine and PBS. Titer was defined by plaque titration assay. Cells inoculated with culture supernatants from uninfected Vero cells mixed with OptiPro serum-free medium supplemented with 0.5% gelatine and PBS, served as mock-infected controls. All infection experiments were carried out under biosafety level three conditions with enhanced respiratory personal protection equipment.

## Plaque titration assay

The amount of infectious virus particles was determined via plaque titration assay. Vero E6 cells were plated at $3.5 \times 10^5$ cell/ml in 24-well and infected with 200 μl of a serial dilution of virus-containing cell-culture supernatant diluted in OptiPro serum-free medium. One hour after adsorption, supernatants were removed and cells overlaid with 2.4% Avicel (FMC BioPolymers) mixed 1:1 in 2x DMEM. Three days postinfection, the overlay was removed, cells were fixed in 6% formaldehyde and stained with 0.2% crystal violet, 2% ethanol, and 10% formaldehyde. Plaque forming units were determined from at least two dilutions for which distinct plaques were detectable.

## Virus exposure of PBMCs

Thirty minutes prior to virus exposure, PBMCs were left mock-treated or treated with Ruxolitinib (10 μM) or Remdesivir (20 μM). Treatment was maintained for the duration of the entire experiment. Virus challenge occurred by inoculation of $0.4 \times 10^6$ cells/ml in RPMI cell-culture medium supplemented with 2% FCS. Four hours postchallenge, cells were centrifuged and supernatants were collected (referred to as inoculum). Cells were resuspended in RPMI cell-culture medium supplemented with 10% FCS and plated at $0.4 \times 10^6$ cell/1.5 ml in 12-wells. In addition, postwash samples were collected. For further sampling, cell-culture supernatant was centrifuged, the supernatant was collected and mixed with OptiPro serum-free medium supplemented with 0.5% gelatine for titration on Vero E6 cell or mixed with RAV1 buffer for viral RNA extraction and stored at −80°C until sample processing. Suspension cells and adherent cells were lysed in Trizol reagents and subjected to total RNA extraction. In PBMC prestimulation experiments, cells were prestimulated for 18 h, stimuli were removed by washing with PBS, and cells were inoculated with SARS-CoV-2 for 24 h as described above. For stimulation, cells were mock-treated or treated with 100 IU/ml IFN-α2a (Roferon), cultured in the presence of supernatants from mock- or SARS-CoV-2-infected Calu-3 cells, either crude or processed by Vivaspin® 20 filtration to obtain the cytokine-containing, but a virus-free fraction of the supernatant, or cultured in the presence of 10% serum collected from three mildly diseased COVID-19 patients (WHO 3; see Appendix Table S1) or healthy control sera. Three hospitalized COVID-19 patients' sera and clinical data were collected at Charité—Universitätsmedizin Berlin in the context of the *PaCOVID-19 Study* (Kurth *et al*, 2020). Patients were recruited between March and November 2020. All patients provided a positive SARS-CoV-2 by RT–PCR from respiratory specimens. The study was approved by the ethics committee of Charité (EA2/066/20). Written informed consent was obtained from all patients or legal representatives.

## Reagents and inhibitors

Ruxolitinib was purchased from InvivoGen and used at 10 μM concentration. Remdesivir (Gilead Sciences) was kindly provided by the Department of Infectious Diseases and Respiratory Medicine, Charité – Universitätsmedizin Berlin. IFN-α2a (Roferon) was obtained from Roche.

## Quantitative Q-RT–PCR

Viral RNA was extracted from cell-culture supernatants using the NucleoSpin RNA virus isolation kit (Macherey-Nagel) according to the manufacturer's instructions. Total RNA extraction from cells and DNase treatment were performed with Direct-zol RNA extraction kit (Zymo Research). Viral genome equivalents were determined using a previously published assay specific for both SARS-CoV and SARS-CoV-2 E gene (Corman et al, 2020). Subgenomic E gene expression was analyzed using the same probe and reverse primer combined with a forward primer, which is located in the SARS-CoV-2 leader region (sgLead-CoV-F: CGA TCT CTT GTA GAT CTG TTC TC; Wölfel et al, 2020). Subgenomic N gene expression was quantified with the following primers and probe: nCoV sgN Fwd: 5'-CGA TCT CTT GTA GAT CTG TTC TC-3', nCoV sgN Rev: 5'-CAG TAT TAT TGG GTA AAC CTT GG-3' and nCoV sgN prb: 5'-56-FAM/ CAG TAA CCA GAA TGG AGA ACG CAG /3BHQ-1-3. To analyze human gene expression, extracted RNA was subjected to cDNA synthesis (NEB, Invitrogen). Quantification of relative mRNA levels was performed with the LightCycler 480 Instrument II (Roche) using Taq-Man PCR technology. For human *IFIT1* and *IFNB1*, a premade primer-probe kit was used (Applied Biosystems, assay IDs: Hs01911452_s1; Hs01077958_s1, respectively). For human *ACE2* (ACE2-F: TGCCTATCC TTCCTATATCAGTCCAA, ACE2-R:GAGTA CAGATTTGTCCAAAATCTAC, ACE2-P: 6-FAM/ATGCCTCCCTGCT CATTTGCTTGGT/IBFQ), *IL-6* (IL-6-F: GGATTCAATGAGGAGACT TGC, IL-6-R: CACAGCTCTGGCTTGTTCC, IL-6-P: 6-FAM/AATCAT CAC/ZEN/TGGTCTTTTGGAGTTTGAGG/IBFQ), and *IFNA1* (IFNA1-F:GGGATGAGGACCTCCTAGACAA, IFNA1-R:CATCACACAGGCTT CCAAGTCA, IFNA1-P:6-FAM/TTCTGCACCGAACTCTACCAGCAGC TG/BHQ), customer-designed oligonucleotides were synthesized by Integrated DNA Technologies (IDT). Relative mRNA levels were determined using the ΔΔCt method using human *RNASEP* (Applied Biosystems) as the internal reference. Data analysis was performed using LightCycler Software 4.1 (Roche).

## Immunoblotting

Cells were washed once with ice-cold PBS and lysed in 60 µl RIPA Lysis Buffer (Thermo Fisher Scientific) supplied with 1% protease inhibitor cocktail set III (Merck Chemicals) for 30 min at 4°C. Cell debris was pelleted for 10 min at 13,000 $g$ and 4°C, and the supernatant was transferred to a fresh tube. Protein concentration was determined with Thermo Scientific's Pierce™ BCA protein assay kit according to the manufacturer's instructions. Protein lysates were mixed with 4 X NuPAGE LDS Sample Buffer (Invitrogen), supplied with 10% 2-mercaptoethanol (Roth), and inactivated for 10 min at 99°C. Proteins were separated by size on a 12% sodium dodecyl sulfate polyacrylamide gel and blotted onto a 0.2 µm PVDF membrane (Thermo Scientific) by semi-dry blotting (BioRad). Human ACE2 was detected using a polyclonal goat anti-human ACE2 antibody (1:500, R&D Systems), a horseradish peroxidase (HRP)-labeled donkey anti-goat antibody (1:5,000, Dinova), and Super Signal West Femto Chemiluminescence Substrate (Thermo Fisher Scientific). As a loading control, samples were analyzed for β-Actin expression using a mouse anti-β-actin antibody (1:5,000, Sigma Aldrich) and an HRP-labeled goat anti-mouse antibody (1:10,000, Dianova).

## HL116 cell-based detection of bioactive IFNs

Cell-culture supernatants of individual cell lines were titrated on HL116 cells that express the luciferase gene under the control of the IFN-inducible 6–16 promoter (Uzé et al, 1994). Cells were PBS-washed, and luciferase expression was quantified using Cell-Culture Lysis Buffer and the Luciferase Assay System (both Promega). The concentration of IFN was quantified using an IFN-α2a (Roferon) standard curve.

## Cytokine profiling

Supernatants from untreated or Ruxolitinib-pretreated and mock-, SARS-CoV-, or SARS-CoV-2-inoculated PBMCs from four donors were collected 48 h postexposure. As a positive control, PBMCs were treated with 1 µg/ml Lipopolysaccharide (LPS, Sigma Aldrich) or 1 µg/ml Phytohaemagglutinin (PHA, Sigma Aldrich) for 48 h. For each condition, samples from four donors were pooled. Cytokines were quantified using a Human/Cytokine/Chemokine/Growth Factor Panel A 48-Plex Premixed Magnetic Bead Multiplex Assay (Merck Millipore), using the Luminex MAGPIX System according to the manufacturer's instructions. Calibration and verification checks were met for all of the analytes. All analytes had standard curves with $R^2$ values > 0.9, except for FGF-2, GM-CSF, IL-9, IL-27, MCP-3, MIP-1β, and PDGF-AA/BB, which had standard curves with $R^2$ values > 0.75.

## Single-Cell RNA-seq

Single-Cell RNA-seq libraries were prepared with the 10× Genomics platform using the Chromium Next GEM Single Cell 3' Reagent Kits v.3.1 following the manufacturer's instructions. Samples were multiplexed using TotalSeq-A Antibodies purchased from BioLegend (A0256, A0258, and A0259). Antibody staining and the subsequent library preparation were performed following the manufacturer's instructions. Quality control of the libraries was performed with the KAPA Library Quantification Kit and Agilent TapeStation. Libraries were sequenced on a HiSeq4000 using the following sequencing mode: read 1: 28 bp, read 2: 91–100 bp, Index i7: 8 bp. The libraries were sequenced to reach ~20,000 reads per cell.

## Single-Cell RNA-seq data analysis

BCL files from the sequencing protocol were processed using the Cell Ranger pipeline v 3.1.0 (10× Genomics) and further analyzed using the Seurat v3.1.4 package (Butler et al, 2018) in R v3.6 (https://www.r-project.org/). Preprocessing of the data was performed using the recommended SCTransform procedure and the IntegrateData with PrepSCTIntegration workflows to eliminate batch effects. A comprehensive description of the code used in the analysis of data is available at https://github.com/GoffinetLab/SARS-CoV-2_PBMC-study. Cell types were identified based on marker gene expression (Schulte-Schrepping et al, 2020): B cells ($CD3D^-$, $MS4A1^+$), CD4$^+$ T cells ($CD3D^+$, $CD8A^-$), CD8$^+$ T cells ($CD3D^+$, $CD8A^+$), NK cells ($CD3D^-$, $CD8A^-$, $NKG7^+$, $GNLY^+$), Monocytes ($CD3D^-$, $CD14^+$, $FCGR3A^+$). Reads aligning to the SARS-CoV or SARS-CoV-2 genome were identified by alignment to

a combined SARS-Cov (AY310120.1, GenBank) and SARS-CoV-2 (NC_045512.2, GenBank) reference using the same Cell Ranger pipeline and visualized in a coverage plot using pyGenomeTracks (Lopez-Delisle *et al*, 2021).

### Cell trajectory analysis

Cell trajectory analysis was performed using the Monocle v2.14.0 package (Trapnell *et al*, 2014) according to the guidelines set out by the developers. Different cell types were subclustered and processed as mentioned above. A resolution parameter of 0.3 was used for clustering. DEGs between clusters were determined using Seurat's FindAllMarkers function (Wilcoxon rank-sum test); of these, genes with a Bonferroni-corrected *P*-value of < 0.05 were imputed as ordering genes to generate the minimum spanning tree using the DDRTree algorithm. Code available at https://github.com/Goffinet Lab/SARS-CoV-2_PBMC-study.

### IFN module score

The IFN-signaling pathway gene set [R-HSA-913531] from the Reactome database (Gillespie *et al*, 2022) was retrieved from the Molecular Signatures Database (MSigDB; Liberzon *et al*, 2015). Cells were scored on their expression of these genes using the AddModuleScore function in Seurat, which is referred to as the IFN module score as the pathway includes genes canonically differentially regulated in response to interferon signaling.

### Flow cytometry analysis

PBS-washed cells were PFA-fixed and immunostained for individual surface protein expression using the following antibodies: Anti-CD3-FITC (#561807; BD Biosciences), anti-CD4-APC (#555349; BD Biosciences), anti-CD14-PE (#561707; BD Biosciences), anti-CD19-FITC (#21270193; ImmunoTools), anti-NRP1/CD304-APC-R700 (#566038, BD Biosciences), anti-PD-1/CD279-PE (#21272794; ImmunoTools), and anti-TIM-3/CD366-FITC (#345022; Biolegend). To determine ACE2 cell surface expression, cells were immunostained with a goat anti-human ACE2 antibody (#AF933, R&D Systems) followed by immunostaining with a secondary antibody donkey anti-goat Alexa Fluor 488 (#A-11055, Thermo Fisher). ACE2-positive HEK293T cells were generated by transduction of cells with retroviral vectors generated by transfection of HEK293T cells with MLV gag-pol (Bartosch *et al*, 2003), pCX4bsrACE2 (Kamitani *et al*, 2006), and pVSV-G (Stewart *et al*, 2003). A FACS Lyric device (Becton Dickinson, Franklin Lakes, NJ, USA) with BD Suite Software was used for analysis.

### Data presentation and statistical analysis

If not stated otherwise, bars show the arithmetic mean of the indicated amount of repetitions. Error bars indicate SEM from the indicated amount of individual experiments. The thumbnail image was generated with Biorender. If not stated otherwise, statistical significance was calculated by performing the Student's *t*-test using GraphPad Prism. *P*-values < 0.05 were considered significant and marked accordingly: $P < 0.05$ (*), $P < 0.01$ (**), or $P < 0.001$ (***); n.s. = not significant ($\geq 0.05$).

## Data availability

The raw sequencing datasets generated during this study are available at the NCBI Gene Expression Omnibus GSE197665 (https://www.ncbi.nlm.nih.gov/geo/query/acc.cgi?acc=GSE197665).

**Expanded View** for this article is available online.

## Acknowledgements

We thank Julian Heinze for excellent technical support. We thank J. S. M. Peiris, University of Hong Kong, China for providing the SARS-CoV isolate HKU-39849. We thank Sandra Pelligrini for the kind gift of the HL116 cell line. We are grateful for access to the BIH Core Facility Sequencing. This work was supported by funding from the Berlin Institute of Health (BIH) to CG. JK is supported by the Center of Infection Biology and Immunity (ZIBI) and Charité PhD Program. Part of this work was supported by Deutsche Forschungsgemeinschaft (DFG) (SFB-TR 84 to CD) and Bundesministerium für Bildung und Forschung (BMBF) through the projects RAPID-II (01KI2006A) to CD. JE and MAM are supported by VW Foundation Grant no. 9A890. A-ES thanks FOR-COVID (Bayerisches Staatsministerium für Wissenschaft und Kunst) and the Helmholtz Association for financial support. Open Access funding enabled and organized by Projekt DEAL.

## Author contributions

**Julia Kazmierski:** Conceptualization; data curation; formal analysis; validation; investigation; visualization; methodology; writing – review and editing. **Kirstin Friedmann:** Conceptualization; formal analysis; validation; investigation; visualization; methodology; writing – review and editing. **Dylan Postmus:** Formal analysis; visualization; writing – review and editing. **Jackson Emanuel:** Formal analysis; investigation; visualization. **Cornelius Fischer:** Data curation; formal analysis. **Jenny Jansen:** Investigation. **Anja Richter:** Investigation. **Laure Bosquillon de Jarcy:** Investigation. **Christiane Schüler:** Investigation. **Madlen Sohn:** Investigation. **Sascha Sauer:** Supervision. **Christian Drosten:** Resources. **Antoine-Emmanuel Saliba:** Resources. **Leif Erik Sander:** Resources. **Marcel A Müller:** Resources; supervision; writing – review and editing. **Daniela Niemeyer:** Conceptualization; data curation; supervision; writing – original draft; writing – review and editing. **Christine Goffinet:** Conceptualization; resources; data curation; formal analysis; supervision; funding acquisition; investigation; visualization; writing – original draft; project administration; writing – review and editing.

In addition to the CRediT author contributions listed above, the contributions in detail are:
JK, KF, DN, and CG conceived and designed the experiments; JK, KF, JE, JJ, AR, LBJ, CS, MS, and DN performed the experiments, JK, KF, DP, JE, CF, LBJ, DN, and CG analyzed the data; CD, A-ES, LES, and MAM provided resources; DN and CG drafted the manuscript; JK, KF, DP, MAM, DN, and CG reviewed and edited the manuscript; SS, DN, and CG supervised the research.

## Disclosure and competing interests statement
Technische Universität Berlin, Freie Universität Berlin and Charité – Universitätsmedizin have filed a patent application for siRNAs inhibiting SARS-CoV-2 replication with DN as co-author. The other authors declare that they have no conflict of interest.

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
