## [Review Process File · Molecular Systems Biology]

Non-productive exposure of PBMCs to SARS-CoV-2 induces cell-intrinsic innate immune responses

Julia Kazmierski, Kirstin Friedmann, Dylan Postmus, Jackson Emanuel, Cornelius Fischer, Jenny Jansen, Anja Richter, Laure Bosquillon de Jarcy, Christiane Schüler, Madlen Sohn, Sascha Sauer, Christian Drosten, Antoine-Emmanuel Saliba, Leif Sander, Marcel Müller, Daniela Niemeyer, and Christine Goffinet

DOI: 10.15252/msb.202210961

Corresponding authors: Christine Goffinet (christine.goffinet@charite.de) , Daniela Niemeyer (daniela.niemeyer@charite.de)

Review Timeline:

Submission Date:	7th Feb 22
Editorial Decision:	7th Mar 22
Revision Received:	5th Jul 22
Editorial Decision:	27th Jul 22
Revision Received:	28th Jul 22
Accepted:	28th Jul 22

Editor: Jingyi Hou

Transaction Report:

7th Mar 2022

Manuscript Number: MSB-2022-10961

Title: Non-productive exposure of PBMCs to SARS-CoV-2 induces cell-intrinsic innate immunity responses

Author: Julia Kazmierski

Kirstin Friedmann

Dylan Postmus

Cornelius Fischer

Jenny Jansen

Anja Richter

Laure Bosquillon de Jarcy

Christiane Schüler

Madlen Sohn

Sascha Sauer

Christian Drosten

Antoine-Emmanuel Saliba

Leif Sander

Daniela Niemeyer

Christine Goffinet

Dear Dr. Goffinet,

Thank you for submitting your work to Molecular Systems Biology. We have now heard back from the three reviewers who agreed to evaluate your manuscript. As you will see from the reports below, the reviewers acknowledge the potential relevance and interest of the presented findings. They raise, however, a series of concerns, which we would ask you to address in a major revision.

I think the reviewers' recommendations are rather clear, so there is no need to reiterate the points listed below. In particular, the reviewers mentioned that additional experiments and analyses would be required to conclusively support the main findings. Further, Reviewer #1 raised concerns about the in vivo relevance of the presented findings (also asked by Reviewer #2), which need to be carefully addressed.

All other issues raised by the reviewers need to be satisfactorily addressed as well. As you may already know, our editorial policy allows in principle a single round of major revision, and it is therefore essential to provide responses to the reviewers' comments that are as complete as possible. Please feel free to contact me in case you would like to discuss in further detail any of the issues raised by the reviewers.

On a more editorial level, we would ask you to address the following issues:

- Please provide a .docx formatted version of the manuscript text (including legends for main figures, EV figures and tables). Please make sure that the changes are highlighted to be clearly visible.
- Please provide individual production quality figure files as .eps, .tif, .jpg (one file per figure).
- Please provide a .docx formatted letter INCLUDING the reviewers' reports and your detailed point-by-point responses to their comments. As part of the EMBO Press transparent editorial process, the point-by-point response is part of the Review Process File (RPF), which will be published alongside your paper.
- Please note that all corresponding authors are required to supply an ORCID ID for their name upon submission of a revised manuscript.
- We replaced Supplementary Information with Expanded View (EV) Figures and Tables that are collapsible/expandable online (see examples in <http://msb.embopress.org/content/11/6/812>). A maximum of 5 EV Figures can be typeset. EV Figures should be cited as 'Figure EV1, Figure EV2' etc... in the text and their respective legends should be included in the main text after the legends of regular figures.

Additional Tables/Datasets should be labeled and referred to as Table EV1, Dataset EV1, etc. Legends have to be provided in a separate tab in case of .xls files. Alternatively, the legend can be supplied as a separate text file (README) and zipped together with the Table/Dataset file.

For the figures and tables that you do NOT wish to display as Expanded View figures, they should be bundled together with their

legends in a single PDF file called *Appendix*, which should start with a short Table of Content. Each legend should be below the corresponding Figure/Table in the Appendix. Appendix figures and tables should be referred to in the main text as: "Appendix Figure S1, Appendix Figure S2, Appendix Table S1" etc. See detailed instructions regarding expanded view here: <https://www.embopress.org/page/journal/17444292/authorguide#expandedview>.

-Before submitting your revision, primary datasets (and computer code, where appropriate) produced in this study need to be deposited in an appropriate public database (see <https://www.embopress.org/page/journal/17444292/authorguide#dataavailability>).

The accession numbers and database should be listed in a formal "Data Availability" section (placed after Materials & Method) that follows the model below (see also <https://www.embopress.org/page/journal/17444292/authorguide#dataavailability>). Please note that the Data Availability Section is restricted to new primary data that are part of this study.

Data availability

- We would encourage you to include the source data for figure panels that show essential quantitative information. Additional information on source data and instruction on how to label the files are available at < <https://www.embopress.org/page/journal/17444292/authorguide#sourcedata> >.

- Our journal encourages inclusion of *data citations in the reference list* to directly cite datasets that were re-used and obtained from public databases. Data citations in the article text are distinct from normal bibliographical citations and should directly link to the database records from which the data can be accessed. In the main text, data citations are formatted as follows: "Data ref: Smith et al, 2001". In the Reference list, data citations must be labeled with "[DATASET]". A data reference must provide the database name, accession number/identifiers and a resolvable link to the landing page from which the data can be accessed at the end of the reference. Further instructions are available at .

- We updated our journal's competing interests policy in January 2022 and request authors to consider both actual and perceived competing interests. Please review the policy <https://www.embopress.org/competing-interests> and update your competing interests if necessary.

Please use the heading "Disclosure statement and competing interests".

- All Materials and Methods need to be described in the main text. We would encourage you to use 'Structured Methods', our new Materials and Methods format. According to this format, the Material and Methods section should include a Reagents and Tools Table (listing key reagents, experimental models, software and relevant equipment and including their sources and relevant identifiers) followed by a Methods and Protocols section in which we encourage the authors to describe their methods using a step-by-step protocol format with bullet points, to facilitate the adoption of the methodologies across labs. More information on how to adhere to this format as well as downloadable templates (.doc or .xls) for the Reagents and Tools Table can be found in our author guidelines: < <https://www.embopress.org/page/journal/17444292/authorguide#researcharticleguide>>. An example of a Method paper with Structured Methods can be found here: .

-Regarding data quantification:

Please ensure to specify the name of the statistical test used to generate error bars and P values, the number (n) of independent experiments (please specify technical or biological replicates) underlying each data point and the test used to calculate p-values in each figure legend. Discussion of statistical methodology can be reported in the materials and methods section, but figure legends should contain a basic description of n, P and the test applied. Graphs must include a description of the bars and the error bars (s.d., s.e.m.). Please also include scale bars in all microscopy images.

- Please provide a "standfirst text" summarizing the study in one or two sentences (approximately 250 characters, including space), three to four "bullet points" highlighting the main findings and a "synopsis image" (550px width and max 400px height, PNG format) to highlight the paper on our homepage.

Here are a couple of examples:

<https://www.embopress.org/doi/10.15252/msb.20199356>

<https://www.embopress.org/doi/10.15252/msb.20209475>
<https://www.embopress.org/doi/10.15252/msb.209495>

When you resubmit your manuscript, please download our CHECKLIST (<https://www.embopress.org/pb-assets/embo-site/EMBO%20Press%20Author%20Checklist-1642513524327.xlsx>) and include the completed form in your submission. *Please note* that the Author Checklist will be published alongside the paper as part of the transparent process (<https://www.embopress.org/page/journal/17444292/authorguide#transparentprocess>).

If you feel you can satisfactorily deal with these points and those listed by the referees, you may wish to submit a revised version of your manuscript. Please attach a covering letter giving details of the way in which you have handled each of the points raised by the referees. A revised manuscript will be once again subject to review and you probably understand that we can give you no guarantee at this stage that the eventual outcome will be favorable.

I look forward to receiving your revised manuscript soon.

Kind regards,
Jingyi

Jingyi Hou
Editor
Molecular Systems Biology

We realize that it is difficult to revise to a specific deadline. In the interest of protecting the conceptual advance provided by the work, we recommend a revision within 3 months (5th Jun 2022). Please discuss the revision progress ahead of this time with the editor if you require more time to complete the revisions. Use the link below to submit your revision:

IMPORTANT: When you send your revision, we will require the following items:

1. the manuscript text in LaTeX, RTF or MS Word format
2. a letter with a detailed description of the changes made in response to the referees. Please specify clearly the exact places in the text (pages and paragraphs) where each change has been made in response to each specific comment given
3. three to four 'bullet points' highlighting the main findings of your study
4. a short 'blurb' text summarizing in two sentences the study (max. 250 characters)
5. a 'thumbnail image' (550px width and max 400px height, Illustrator, PowerPoint or jpeg format), which can be used as 'visual title' for the synopsis section of your paper.
6. Please include an author contributions statement after the Acknowledgements section (see <https://www.embopress.org/page/journal/17444292/authorguide>)
7. Please complete the CHECKLIST available at (<https://bit.ly/EMBOPressAuthorChecklist>).

Please note that the Author Checklist will be published alongside the paper as part of the transparent process (<https://www.embopress.org/page/journal/17444292/authorguide#transparentprocess>).

See also figure legend guidelines: <https://www.embopress.org/page/journal/17444292/authorguide#figureformat>

9. Please note that corresponding authors are required to supply an ORCID ID for their name upon submission of a revised manuscript (EMBO Press signed a joint statement to encourage ORCID adoption).

(<https://www.embopress.org/page/journal/17444292/authorguide#editorialprocess>)

Currently, our records indicate that the ORCID for your account is 0000-0002-3959-004X.

Link Not Available

The system will prompt you to fill in your funding and payment information. This will allow Wiley to send you a quote for the article processing charge (APC) in case of acceptance. This quote takes into account any reduction or fee waivers that you may be eligible for. Authors do not need to pay any fees before their manuscript is accepted and transferred to the publisher.

*** PLEASE NOTE *** As part of the EMBO Press transparent editorial process initiative (see our Editorial at <https://dx.doi.org/10.1038/msb.2010.72>), Molecular Systems Biology publishes online a Review Process File with each accepted manuscripts. This file will be published in conjunction with your paper and will include the anonymous referee reports, your point-by-point response and all pertinent correspondence relating to the manuscript. If you do NOT want this File to be published, please inform the editorial office at msb@embo.org within 14 days upon receipt of the present letter.

Reviewer #1:

Within this manuscript, Kazmierski and Friedmann et al. analysed the effects of SARS-CoV and SARS-CoV-2 on ex vivo exposed healthy and unstimulated PBMCs. They show that infection of PBMCs with both viruses lead to a non-productive infection, with no viral particles released whereas a small percentage of cells are viral RNA positive. They performed bulk and single cell RNA sequencing analysis revealing an induction of interferon-stimulated genes, which was most pronounced in monocytes. These effects were not observed for SARS-CoV infected PBMCs. SARS-CoV-2-RNA-positive and negative monocytes showed no differences in their expression profiles. IFN module score was elevated in SARS-CoV-2-RNA-negative monocytes.

The MS is timely and relevant and reports an interesting finding important to the research community. The reviewer acknowledges the amount of work performed, in particular working with BSL3 pathogens. Nevertheless, the in vivo relevance of the observed effects is not fully clear. Therefore, the study needs more experimental evidence to support the hypothesis. In general, it is unlikely that fully naïve, unstimulated peripheral blood cells are exposed to SARS-CoV-2 in vivo. This should occur exclusively in tissue invading cells. However, these cells must have been pre-activated at least by chemotaxis inducing triggers. This could be addressed experimentally.

Major points:

-What is the in vivo relevance of this ex vivo study? Can the authors detect similar expression profiles in PBMCs derived from mild COVID-19 patients? This comparison would be important to support the conclusions drawn in the paper. In general, an extended critical discussion of the in vivo relevance would be helpful. There is a multitude of publications dealing with single cell RNAseq from severe versus mild COVID-19 disease.

<https://www.nature.com/articles/s41591-021-01329-2>

<https://www.nature.com/articles/s41421-020-00225-2>

-Does the cytokine profile of infected PBMCs change, when compared to Mock-infected? The authors should perform a cytokine profiling of pro-inflammatory cytokines and type I interferons in the supernatant of infected and uninfected PBMCs. This is of particular importance for two reasons: i) to correlate sequencing analysis with an additional ex vivo analysis, especially in regard to the findings on interferon signalling ii) to correlate the data to the in vivo situations, where several plasma cytokines are elevated. This should also be performed with SARS-CoV!

Ruxolitinib treatment may also provide further information here.

-What happens in pre-stimulated or diseased PBMCs? Is the response the same as compared to unstimulated and healthy PBMCs? These experiments are important to assess the in vivo relevance, since severe COVID-19 often occurs in patients with underlying chronic diseases, where PBMCs are under constant stimulation. One could repeat the experiments partly with diabetic or obese patient derived cells. The experiment could also be performed in healthy donor cells with a pre-stimulation step that mimics this situation, e.g. by using associated DAMPs.

-in the same direction: the finding that there is less viral RNA found in monocytes with an interferon stimulated gene-signature (ISG) should be experimentally addressed. E.g. by stimulating cells with type I or type II interferons and testing for viral RNA content. If the conclusion drawn by the authors is correct, there should be less viral RNA in these cells, compared to unstimulated control cells. This may even have therapeutic and translational relevance. In the end, this would confirm the importance of interferon dependent stimulation (for example via a vaccine) for innate immune signalling in virus infected monocytes.

-Critical discussion of the SARS-CoV data. Why does all the sequencing and in vitro data look similar or even identical to the Mock controls (Fig. 4E)? Are there really no changes in the gene expression signatures, which if true is somehow surprising, when viral RNA is found?! There are published data of SARS-CoV exposed monocytes from the early 2000 years. SARS-CoV should induce at least some cytokine signalling. Please discuss literature:

https://www.liebertpub.com/doi/10.1089/vim.2011.0099?url_ver=Z39.88-2003&rfr_id=ori:rid:crossref.org&rfr_dat=cr_pub%20%20pubmed

<https://www.ncbi.nlm.nih.gov/pmc/articles/PMC1143636/>

-According to the literature, other coronaviruses such as 229E strongly induce interferon signalling in infected macrophages. Yet

their clinical features in comparison to SARS-CoV-2 are clearly different. Please discuss.

Minor points:

- Statistical analysis in Figure 2A missing. Required to see if the level of IFIT1 upon Ruxolitinib is significantly reduced. This is of particular importance for the statements drawn in the paper.
- Description of the statistical analysis in the figure legends is missing. Please indicate. In addition, the reviewer believes that statistical analyses were not always be indicated. For instance in Figure 1C.
- Figure 2B. For some bar graphs, some data points are missing. According to the legends n=4 was performed, however, this is not always depicted. Please clarify.
- Description of the biological replications in the figure legends is sometimes missing. Please indicate.
- Discussion of the CD163 data. Here also pro-inflammatory cytokines seem to be important (IL-1 for instance), which stands in contrast to the conclusions drawn. Please discuss these results in comparison to the rest of the data and what that means to the in vivo relevance?
- line 42: add (ISG) behind interferon stimulated gene

Reviewer #2:

In this study, the authors perform SARS-CoV-2 ex vivo infections on PBMCs, with the goal of determining if these can be productively infected and to characterize the cellular responses to infectious virions. The study asks an important question in relation to the role of PBMCs upon infection, how they could (or not) be involved in virus dissemination and how they may contribute to the adequate overall innate immune response for infection control. The study would appeal to a broad audience, including virologists, immunologists, and clinicians.

In this work the authors performed ex vivo infections, followed by direct virus titration at different timepoints, quantification of virus RNA, and single cell RNA sequencing to thoroughly determine which cells are mainly engaged, and characterize the response of these cells and bystander ones. The main finding is that while PBMCs do not support productive replication of SARS-CoV-2, virus material remains attached or is internalized by these cells in an ACE2 independent fashion. This is an important finding since there has been controversy in relation to the permissibility of PBMCs to SARS-CoV-2 infection. They also find that upon detection, cells -mainly monocytes- mount robust innate immune responses engaging IRF3 signaling and a strong induction of ISGs, with no engagement of NFkB pathway or production of cytokines.

The study is well planned and executed, and the conclusions are justified. My main criticism of the work is it lacks a bit of novelty, and that the authors do not go deeply into the mechanism of how virus RNA/material gets internalized or attached.

Major points

- Fig. 1: I believe it would be useful to understand the nature of the virus-cell interaction, so the authors could try to show either infectious viruses inside cells or viral antigen/protein. I understand showing infectious virus inside cells can be technically challenging, but maybe they could try staining for N protein or western blot? To get rid of potential viruses attached to the outside of the cells they could maybe try enzymatic treatment like trypsin.
- Fig. 1E: After establishing there is no productive infection in PBMCs, I believe that if they were trying to determine if there is replication of the genome, they should try earlier time points. Early stages of the virus life cycle occur only hours after entering the cell. Also, they should include post wash RNA levels for adherent and suspension cells.
- Fig. 1C and 1D.: why is there an increase in viral RNA in Fig. 1C at post wash but the opposite is observed in Fig. 1D? This also doesn't match the findings in Fig. 1A, where there are less pfu in the post wash condition and since there is no productive infection then the viral RNA should decrease instead of increase as seen in 1C.
- Though not a necessary requirement, it would improve the manuscript if the authors could obtain PBMCs from infected patients (mild cases), perform scRNASeq, run the same analyses and compare them. I understand this may not be possible and that even if it is, most likely you would be comparing different time points.
- If the authors would want to expand their observations of the interactions between PBMCs, infectious virions and signaling (mainly cytokines) from productively infected epithelial cells, they could try to infect Calu-3 cells with SARS-CoV-2, collect the supernatant after 48 hours and expose fresh PBMCs to it. They could do this in two conditions, one with supernatant containing cytokines and infectious virus, and the other one containing only cytokines after treatment with a detergent at low concentration (ex: Triton) that would kill the virus but not the cytokines.

Minor points

- Title: Innate immune responses instead of innate immunity responses?
- Line 62: define manifestation index
- Line 72: the authors may want to add paloxavir now
- Fig. 1: In the figure, the authors sometimes use the word "mock" to refer to uninfected, untreated cells and sometimes to refer to untreated, SARS-CoV-2 infected cells. This is confusing at times.
- Fig. 1A and 1B: why do the authors not show the data for timepoints earlier than 72 hpe?

- Line 117 - 118: In relation to previous comment, the authors talk about infectivity at 24 hours but this datapoint is not shown?
- the authors may want to address why in Fig. 1A there is no virus at 72 hpe but there is plenty of virus at 72 hpe in Fig. 1B?
- Line 256: remove "in line with"
- Line 260-263: homogenize font size
- Line 430: remove "extraction"
- Line 456 - 458: homogenize font
- The reference for Corman et al., 2020 is missing

Reviewer #3:

This manuscript describes an ex vivo study of SARS-CoV and SARS-CoV-2 infection in PBMCs, which is important to the understanding of the host cell spectrum of Sarbecoviruses and evaluating the roles of virus+ myeloid cells observed in clinical samples. The manuscript is well written. I have three concerns about the data analysis.

1. The authors should analyze the detectability of different ORFs in PBMCs. SARS-CoV-2 RNAs can be detected in myeloid cells derived from clinical samples and demonstrated a gradient from the 5' end to 3' end along the viral genome, indicating subgenomic transcription. I am curious whether ex vivo studies could reproduce this observation.
2. One conclusion of this study is that physical interaction and internalization of SARS-CoV-2 with monocytes can induce interferon responses. I am curious what components of the viral particles are important to induce the interferon responses of monocytes and whether internalization is necessary. The authors are expected to discuss whether cytokine stimulations could sensitize the internalization of myeloid cells so that SARS-CoV-2 RNA could be detected more frequently in macrophages and neutrophils.
3. The current trajectory analysis is weird. SARS-CoV treated samples should not be mixed with SARS-CoV-2 treated samples to infer potential developmental trajectories.

Overall, this study provides an ex vivo study of SARS-CoV and SARS-CoV-2 infection in PBMCs, which is insightful and important to understand the biology of SARS-CoV-2.

Reviewer #1:

Within this manuscript, Kazmierski and Friedmann et al. analysed the effects of SARS-CoV and SARS-CoV-2 on ex vivo exposed healthy and unstimulated PBMCs. They show that infection of PBMCs with both viruses lead to a non-productive infection, with no viral particles released whereas a small percentage of cells are viral RNA positive. They performed bulk and single cell RNA sequencing analysis revealing an induction of interferon-stimulated genes, which was most pronounced in monocytes. These effects were not observed for SARS-CoV infected PBMCs. SARS-CoV-2-RNA-positive and negative monocytes showed no differences in their expression profiles. IFN module score was elevated in SARS-CoV-2-RNA-negative monocytes.

The MS is timely and relevant and reports an interesting finding important to the research community. The reviewer acknowledges the amount of work performed, in particular working with BSL3 pathogens. Nevertheless, the in vivo relevance of the observed effects is not fully clear. Therefore, the study needs more experimental evidence to support the hypothesis. In general, it is unlikely that fully naïve, unstimulated peripheral blood cells are exposed to SARS-CoV-2 in vivo. This should occur exclusively in tissue invading cells. However, these cells must have been pre-activated at least by chemotaxis inducing triggers. This could be addressed experimentally.

Major points:

-What is the in vivo relevance of this ex vivo study? Can the authors detect similar expression profiles in PBMCs derived from mild COVID-19 patients? This comparison would be important to support the conclusions drawn in the paper. In general, an extended critical discussion of the in vivo relevance would be helpful. There is a multitude of publications dealing with single cell RNAseq from severe versus mild COVID-19 disease.

<https://www.nature.com/articles/s41591-021-01329-2>

<https://www.nature.com/articles/s41421-020-00225-2>

Reply: We now added a paragraph highlighting to which extent our findings align to data obtained from COVID-19 patients' PBMCs:

“This observation is well in line with ex vivo data from PBMCs derived from COVID-19 patients showing highest, monocyte-specific induction of IFN-mediated responses that are inversely proportional to the degree of disease (Stephenson et al. 2021; Schulte-Schrepping et al. 2020; Arunachalam et al. 2020), as well as absence of proinflammatory cytokine expression (Stephenson et al. 2021; G. Xu et al. 2020; Arunachalam et al. 2020). Because our cellular model lacks the complexity of interactions occurring in vivo between circulating immune cells and tissue-resident cells, including those of the productively infected respiratory tract, we hypothesize that it approaches the situation in mildly infected individuals with a transient, rapidly controlled phase of virus replication involving a limited amount of virus-induced cell damage and immune dysregulation.” (Lines 387-397)

-Does the cytokine profile of infected PBMCs change, when compared to Mock-infected? The authors should perform a cytokine profiling of pro-inflammatory cytokines and type I interferons in the supernatant of infected and uninfected PBMCs. This is of particular importance for two reasons: i) to correlate sequencing analysis with an additional ex vivo analysis, especially in regard to the findings on interferon signalling ii) to correlate the data to the in vivo situations, where several plasma cytokines are elevated. This should also be performed with SARS-CoV!

Ruxolitinib treatment may also provide further information here.

Reply: Thank you for this suggestion. We have conducted an extensive quantitative cytokine profiling analysis using supernatants from PBMCs exposed to SARS-CoV, SARS-CoV-2 in the presence and absence of Ruxolitinib (new **Figure 2C** and new **Appendix Figure S3**). In addition, we tested for bioactive type I IFN in corresponding supernatants (new **Figure 2D**). These new data demonstrate secretion of IFN-alpha and ISG-encoded cytokines, including IP-10/CXCL10, MCP-1/CCL2 and MCP-3/CCL7 in PBMC cultures exposed to SARS-CoV-2, and to a lower extent, to SARS-CoV, and underline the absence of pro-inflammatory cytokine induction observed on the transcriptomic level. This IFN signature at the level of secreted protein was accompanied by release of bioactive type I IFN by SARS-CoV-2-exposed PBMCs. We added the following text passage describing the new results:

*“However, IFN- α 2 and IFN-stimulated IP-10, MCP-1 and MCP-3 proteins, as opposed to IL-6 and several other cytokines (**Appendix Fig. S3**) were secreted in the supernatant of exposed PBMCs in a Ruxolitinib-sensitive manner, with overall higher levels in SARS-CoV-2- than in SARS-CoV-exposed cultures (**Fig. 2C**). Furthermore, bioactive IFN was detected in the supernatant of corresponding cultures, with highest levels upon SARS-CoV-2 exposure, while SARS-CoV inoculation induced the release of bioactive IFN at inferior levels that were statistically indistinguishable from those in mock-exposed cultures (**Fig. 2D**).”* (Lines 184-191)

-What happens in pre-stimulated or diseased PBMCs? Is the response the same as compared to unstimulated and healthy PBMCs? These experiments are important to assess the in vivo relevance, since severe COVID-19 often occurs in patients with underlying chronic diseases, where PBMCs are under constant stimulation. One could repeat the experiments partly with diabetic or obese patient derived cells. The experiment could also be performed in healthy donor cells with a pre-stimulation step that mimics this situation, e.g. by using associated DAMPs.

Reply: Thank you for this interesting idea that is shared with Reviewers #2 and #3.. We performed a series of additional experiments addressing this comment, with the corresponding data summarized in new **Figure EV5**. To mimic the environment of circulating PBMCs of a SARS-CoV-2-infected individual, we individually pre-treated PBMCs with type I IFN, with supernatant from SARS-CoV-2-infected lung epithelial cell cultures, and with serum obtained at an early stage post-infection from mildly COVID-19 diseased individuals. The two latter treatments, but not type I IFN treatment, statistically significantly enhanced the cell-associated quantities of viral RNA upon exposure, suggesting cytokines induced by SARS-CoV-2 infection in vitro and in vivo may favor interaction of PBMCs with virus particles. However, the two treatments did not modulate cellular gene expression to detectable levels,

while IFN, as expected, upregulated *IFIT1* mRNA expression. Together, these new results suggest that PBMCs that are “primed” by stimulation with a cytokine milieu that is characteristic of an ongoing systemic SARS-CoV-2 infection display a slightly increased ability to physically interact with viral particles, in the absence of detectable changes of *IFIT1* and *IL6* mRNA expression:

*“Finally, we were intrigued whether pre-activation would result in an altered ability of PBMCs to interact with, internalize and sense SARS-CoV-2, as opposed to freshly isolated PBMCs from healthy donors. To mimic the environment of circulating PBMCs of a SARS-CoV-2-infected individual, we individually pre-treated PBMCs with type I IFN, with supernatant from SARS-CoV-2-infected lung epithelial cell cultures, and with serum obtained at an early stage post-infection from mildly COVID-19 diseased individuals (Fig. EV5). IFN- α 2a pre-treatment did not influence the amount of detectable viral RNA, neither in the cell-associated fraction (Fig. EV5A, left panel) nor in the supernatant (Fig. EV5B, left panel), despite expected IFN-mediated enhancement of *IFIT1* (Fig. EV5C) but not *IL6* mRNA (Fig. EV5D) expression.*

*Pre-treatment of PBMCs with virus-free, cytokine-containing supernatant derived from SARS-CoV-2-infected Calu-3 cell cultures resulted in a mild (1.6-fold), but statistically significant increase of cell-associated viral RNA copies (Fig. EV5B, middle panel) when compared to cells stimulated with supernatant from uninfected Calu-3 cells, in the absence of changes concerning viral RNA quantities (Fig. EV5A, middle panel) in the culture supernatant, as well as *IFIT1* and *IL6* mRNA expression (Fig. EV5C-D, middle panels). Similarly, cultivation of PBMCs in the presence of serum from COVID-19 patients, as opposed to mock or control serum treatment prior to SARS-CoV-2 exposure, was followed by an 1.8-fold higher abundance of viral genomic RNA in the cellular fraction (Fig. EV5B, right panel). In summary, these results suggest that PBMCs that are “primed” by stimulation with a cytokine milieu that is characteristic of an ongoing systemic SARS-CoV-2 infection display a slightly increased ability to physically interact with viral particles, in the absence of detectable changes of *IFIT1* and *IL6* mRNA expression.” (Lines 289-310)*

“Interestingly, pre-stimulation of PBMCs in a cytokine-containing milieu using virus-free supernatants from infected lung epithelial cell cultures and sera from COVID-19 patients sensitized cells for a slightly more efficient uptake of particles.” (Lines 337-339)

-in the same direction: the finding that there is less viral RNA found in monocytes with an interferon stimulated gene-signature (ISG) should be experimentally addressed. E.g. by stimulating cells with type I or type II interferons and testing for viral RNA content. If the conclusion drawn by the authors is correct, there should be less viral RNA in these cells, compared to unstimulated control cells. This may even have therapeutic and translational relevance. In the end, this would confirm the importance of interferon dependent stimulation (for example via a vaccine) for innate immune signalling in virus infected monocytes.

Reply: Our favored model concerning the relationship between the IFN-stimulated gene signature (illustrated by the IFN module score) and the status of SARS-CoV-2 RNA at the single cell level is as follows: In the absence of productive replication, sensing of viral

PAMPs in cells which internalized virion particles (identified by detectable viral reads) is largely dampened by the delivery of virion-packaged antagonists, whereas cells internalizing virions at a level below our detection limit remain sensing-competent and alert bystander cells, resulting in an elevated IFN module score in the latter. A potentially additional phenomenon is that virion-packaged antagonists lower the overall base-line IFN/ISG level in invaded cells, *per se* resulting in a comparably elevated IFN module score in bystander cells. We favor this model because in our experimental set-up, we identified changes of the IFN module score at the single cell level as a consequence of virus exposure. Testing the inverse scenario, which constitutes an IFN-imposed modulation of virus particle uptake and of intracellular presence of viral RNA at the level of individual cells, would have required another set of laborious and expensive experiments at the level of single cell RNA-sequencing, which were beyond the scope of this study. Furthermore, this scenario would imply the existence of an essential, IFN-sensitive step in the uptake of virions which we deem unlikely given the unspecific and the non-replicative nature of the particle uptake. Nevertheless, we partially addressed this idea using a bulk approach (new **Figure EV5**), that however failed to reveal changed quantities of viral RNA uptake upon IFN stimulation. To integrate this topic, the following text passages were added/expanded to the discussion:

“Bystander cells displayed enhanced ISG expression, suggesting that sensing of viral PAMPs in cells which internalized virion particles (identified by detectable viral reads) is largely dampened by the delivery of virion-packaged antagonists, whereas cells internalizing virions at a level below our detection limit remain sensing-competent and alert bystander cells, resulting in an elevated IFN module score in the latter. A potentially additional phenomenon is that virion-packaged antagonists lower the overall base-line IFN/ISG level in invaded cells, conversely resulting in a comparably elevated IFN module score in bystander cells. We favor this model because in our experimental set-up, we identified changes of the IFN module score at the single cell level as a consequence of virus exposure. Interestingly, and in analogy to our findings, uptake of SARS-CoV by CD14-purified monocytes was found to correlate with a low-to-absent base-line level of IFN expression (Yilla et al. 2005). However, virus preparations were not cleared from contaminating cytokines originating from the producer cell during virus stock production, and whether the low IFN state was a cause or consequence of SARS-CoV-2 exposure was not investigated in that particular study (Yilla et al. 2005), making it difficult to draw analogies to our findings. Indeed, our data cannot exclude the potentially additionally contributing reciprocal scenario of a more efficient and more probable internalization specifically into cells with a low ISG profile, which however would imply the existence of an essential, IFN-sensitive step in the uptake of virions which we deem unlikely given the receptor-independent uptake and the non-reproductive nature of the particle uptake. Indeed, IFN treatment prior to SARS-CoV-2 exposure failed to change quantities of viral RNA uptake upon IFN stimulation as judged in a bulk approach.” (Lines 416-436)

-Critical discussion of the SARS-CoV data. Why does all the sequencing and in vitro data look similar or even identical to the Mock controls (Fig. 4E)? Are there really no changes in the gene expression signatures, which if true is somehow surprising, when viral RNA is found?! There are published data of SARS-CoV exposed monocytes from the early 2000 years. SARS-CoV should induce at least some cytokine signalling. Please discuss literature:

https://www.liebertpub.com/doi/10.1089/vim.2011.0099?url_ver=Z39.88-2003&rft_id=ori:rid:crossref.org&rft_dat=cr_pub%20%20pubmed

<https://www.ncbi.nlm.nih.gov/pmc/articles/PMC1143636/>

Reply: We agree that this topic is worth to be deepened in our study. Our additional experiments (new **Figure 2C-D**) which now complement our RNA datasets show that, even if at inferior levels when compared to SARS-CoV-2, SARS-CoV exposure can trigger secretion of IFN- α and IFN-induced cytokines at the protein level, as well as small quantity of bioactive IFN, even if the latter was statistically indistinguishable from supernatants of mock-exposed cells. We assume that these findings can be explained by the more stable nature of protein as compared to mRNA. Therefore, based on our new additional data, we softened our initially rather categorical exclusion of gene expression changes upon SARS-CoV exposure in the abstract and the title of Figure 2 as follows:

“Together, non-productive physical interaction of PBMCs with SARS-CoV-2-, and to a much lesser extent, SARS-CoV particles stimulates JAK/STAT-dependent, monocyte-accentuated innate immune responses” (Lines 61-63)

Exposure of PBMCs to SARS-CoV-2, and to a much lower extent SARS-CoV, triggers a JAK/STAT-dependent cell-intrinsic innate immune response (Lines 167-168)

*“However, IFN- α 2 and IFN-stimulated IP-10, MCP-1 and MCP-3 proteins, as opposed to IL-6 and several other cytokines (**Appendix Fig. S3**) were secreted in the supernatant of exposed PBMCs in a Ruxolitinib-sensitive manner, with overall higher levels in SARS-CoV-2- than in SARS-CoV-exposed cultures (**Fig. 2C**). Furthermore, bioactive IFN was detected in the supernatant of corresponding cultures, with highest levels upon SARS-CoV-2 exposure, while SARS-CoV inoculation induced the release of bioactive IFN at inferior levels that were statistically indistinguishable from those in mock-exposed cultures (**Fig. 2D**).” (Lines 184-191)*

The two publications mentioned by the reviewer describe gene expression changes induced by SARS-CoV exposure in DC-SIGN-transfected THP-1 cells and monocyte-derived, fully differentiated macrophages, which we believe each are quite different to the primary, non-differentiated monocytes in the context of PBMCs used in our study. We now discuss one study that investigated SARS-CoV uptake and gene expression changes upon virus exposure in CD14-purified monocytes, thereby being better suited for contextualization:

“Another report suggested inefficient de novo SARS-CoV production in CD14-purified monocytes, despite detectable virus particle uptake, presumably through phagocytosis (Yilla et al. 2005).” (Lines 325-327)

“Interestingly, and in analogy to our findings, uptake of SARS-CoV by CD14-purified monocytes was found to correlate with a low-to-absent base-line level of IFN expression (Yilla et al. 2005). However, virus preparations were not cleared from contaminating cytokines originating from the producer cell during virus stock production, and whether the low IFN state was a cause or consequence of SARS-CoV-

2 exposure was not investigated in that particular study (Yilla et al. 2005), making it difficult to draw analogies to our findings.” (Lines 424-430)

-According to the literature, other coronaviruses such as 229E strongly induce interferon signalling in infected macrophages. Yet their clinical features in comparison to SARS-CoV-2 are clearly different. Please discuss.

Reply: We have now included a paragraph on the differences between other human CoVs compared to SARS-CoV and SARS-CoV-2 in regards to innate immune induction and replication in monocytes and macrophages:

“In contrast, endemic human CoVs, including 229E, have been shown to actively enter and replicate in blood-derived monocytic cells and macrophages (Funk et al. 2012; Desforges et al. 2007), in line with detectable expression of the cellular 229E-specific receptor CD13/APN (Yeager et al. 1992; Funk et al. 2012) and triggering a strong infection-induced type I interferon responses in the monocytic cell compartments (Cheung et al. 2005; Desforges et al. 2007). In contrast to 229E, ex vivo exposure of monocytes or macrophages to SARS-CoV-2 triggers a type I IFN-dependent response in the absence of productive infection (this manuscript; (Zankharia et al. 2022; Zheng et al. 2020), however, in vivo studies clearly demonstrate the contribution of monocytes and macrophages to the SARS-CoV-2-induced disease progression as a consequence of the constant exposure to cytokines and viral PAMPs, eventually resulting in a gradually increasing dysregulated myeloid cell compartment ((Kosyreva et al. 2021; Schulte-Schrepping et al. 2020; Leon et al. 2022). In contrast, the high IFN induction of the low pathogenic HCoV 229E early upon infection, is thought to be beneficial for a rapid, immune-mediated viral clearance, whereas the highly pathogenic HCoVs SARS-CoV and SARS-CoV-2 encode numerous viral antagonists to evade innate signaling, eventually resulting in blunted activation of the host cellular immunity and delayed viral clearance in vivo (Fung and Liu 2019; Kim and Shin 2021).“ (Lines 450-466)

Minor points:

-Statistical analysis in Figure 2A missing. Required to see if the level of IFIT1 upon Ruxolitinib is significantly reduced. This is of particular importance for the statements drawn in the paper.

Reply: We added the statistical analysis to Figure 2A, which yielded statistical significant differences regarding *IFIT1* mRNA expression between cells in the presence and absence of Ruxolitinib treatment at time points 16, 24, 48 hours (adherent cells) and 16, 24, 48, and 144 hours (suspension cells) post-exposure to SARS-CoV-2.

- Description of the statistical analysis in the figure legends is missing. Please indicate. In addition, the reviewer believes that statistical analyses were not always be indicated. For instance in Figure 1C.

Reply: Throughout the paper, otherwise indicated explicitly in the legend, nonsignificant p values are not shown. We would like to apologize for the flaw in the original Figure 1C that has been corrected and now does not require statistical analysis for interpretation.

-Figure 2B. For some bar graphs, some data points are missing. According to the legends n=4 was performed, however, this is not always depicted. Please clarify.

Reply: For some samples from the adherent fraction shown in Figure 2A and 2B, the housekeeping gene was undetectable and respective samples were excluded.

-Description of the biological replications in the figure legends is sometimes missing. Please indicate.

Reply: We now indicate the information on biological replicates in each figure legend.

-Discussion of the CD163 data. Here also pro-inflammatory cytokines seem to be important (IL-1 for instance), which stands in contrast to the conclusions drawn. Please discuss these results in comparison to the rest of the data and what that means to the in vivo relevance?

Reply: We added a paragraph to discuss our data in light of the existing literature on CD163 in SARS-CoV-2 infection:

“The literature suggests that CD163 expression in monocytes and macrophages is tightly regulated by pro- and anti-inflammatory cytokines (Reviewed in (Etzerodt and Moestrup 2013)), but is also inducible following Toll like receptor (TLR)-signaling, indicating multiple mechanisms of CD163 regulation (Weaver et al. 2007). Our ex vivo exposure approach does not mirror the cytokine-containing environment monocytes and macrophages of SARS-CoV-2-infected individuals are exposed to, which might, among other reasons, explain the overall mild induction of CD163 in this cellular compartment observed in our data as opposed to data from COVID-19 patients (Gómez-Rial et al. 2020; Wendisch et al. 2021). Our single cell RNA-sequencing dataset, however, allows us to speculate about another potential layer of SARS-CoV-2-induced CD163 up-regulation in monocytes in the absence of pro-inflammatory cytokines as a consequence of virus uptake and innate sensing of viral compartments through pattern recognition receptors, such as TLRs.” (Lines 404-415)

-line 42: add (ISG) behind interferon stimulated gene

Reply: we added the abbreviation.

Reviewer #2:

In this study, the authors perform SARS-CoV-2 ex vivo infections on PBMCs, with the goal of determining if these can be productively infected and to characterize the cellular responses to infectious virions. The study asks an important question in relation to the role of PBMCs upon infection, how they could (or not) be involved in virus dissemination and how they may contribute to the adequate overall innate immune response for infection control. The study would appeal to a broad audience, including virologists, immunologists, and clinicians.

In this work the authors performed ex vivo infections, followed by direct virus titration at different timepoints, quantification of virus RNA, and single cell RNA sequencing to thoroughly determine which cells are mainly engaged, and characterize the response of these cells and bystander ones. The main finding is that while PBMCs do not support productive replication

of SARS-CoV-2, virus material remains attached or is internalized by these cells in an ACE2 independent fashion. This is an important finding since there has been controversy in relation to the permissibility of PBMCs to SARS-CoV-2 infection. They also find that upon detection, cells -mainly monocytes- mount robust innate immune responses engaging IRF3 signaling and a strong induction of ISGs, with no engagement of NFkB pathway or production of cytokines.

The study is well planned and executed, and the conclusions are justified. My main criticism of the work is it lacks a bit of novelty, and that the authors do not go deeply into the mechanism of how virus RNA/material gets internalized or attached.

Major points

- Fig. 1: I believe it would be useful to understand the nature of the virus-cell interaction, so the authors could try to show either infectious viruses inside cells or viral antigen/protein. I understand showing infectious virus inside cells can be technically challenging, but maybe they could try staining for N protein or western blot? To get rid of potential viruses attached to the outside of the cells they could maybe try enzymatic treatment like trypsin.

Reply: Thank you for this suggestion. In an attempt to address it, we MACS-purified CD14-positive monocytes via negative selection and challenged them with SARS-CoV-2 (see figure below for the reviewer's discretion). However, positivity based on the percentage and the mean fluorescence intensity (MFI) upon intracellular N-staining was very close to background levels and too low to enable us to conduct functional assays aiming at elucidating virus cell interaction parameters. In contrast, specificity and sensitivity of the antibody was confirmed on parallelly infected Calu-3 cells.

CD14⁺ Monocytes were isolated from PBMCs and exposed to SARS-CoV-2 for 24 hours. Cells were PFA-fixed and stained intracellularly for SARS-CoV-2 N protein (Rabbit-anti SARS-CoV-2 Nucleocapsid, Biozol, #GTX635680) and subjected to flow cytometry analysis. SARS-CoV-2-infected Calu-3 cells were used as positive control.

- Fig. 1E: After establishing there is no productive infection in PBMCs, I believe that if they were trying to determine if there is replication of the genome, they should try earlier time points. Early stages of the virus life cycle occur only hours after entering the cell. Also, they should include post wash RNA levels for adherent and suspension cells.

Reply: We now included earlier time points (8, 12, 24, 48 hours) for detection of infectivity in the supernatant (**Fig. 1A**), viral RNA in the supernatant (**Fig. 1C**), cell-associated viral RNA (**Fig. 1E**) as well as cell-associated subgenomic (sgRNA) E and N RNA (**EV Fig. 1A-B**). For the early time points, up to 48 hours, we decided to pool suspension and adherent cell fractions to maximize the RT-Q-PCR sensitivity for detection of cell-associated RNA and sgRNA. The newly generated data support our finding of the absence of SARS-CoV and SARS-CoV-2 RNA replication in human PBMCs.

- Fig. 1C and 1D.: why is there an increase in viral RNA in Fig. 1C at post wash but the opposite is observed in Fig. 1D? This also doesn't match the findings in Fig. 1A, where there are less pfu in the post wash condition and since there is no productive infection then the viral RNA should decrease instead of increase as seen in 1C.

Reply: We would like to apologize for the flaw in the original Figure 1C that has been corrected.

- Though not a necessary requirement, it would improve the manuscript if the authors could obtain PBMCs from infected patients (mild cases), perform scRNASeq, run the same analyses and compare them. I understand this may not be possible and that even if it is, most likely you would be comparing different time points.

Reply: We now added a paragraph highlighting to which extent our findings align to published data obtained from COVID-19 patients' PBMCs:

“This observation is well in line with ex vivo data from PBMCs derived from COVID-19 patients showing highest, monocyte-specific induction of IFN-mediated responses that are inversely proportional to the degree of disease (Stephenson et al. 2021; Schulte-Schrepping et al. 2020; Arunachalam et al. 2020), as well as absence of proinflammatory cytokine expression (Stephenson et al. 2021; G. Xu et al. 2020; Arunachalam et al. 2020). Because our cellular model lacks the complexity of interactions occurring in vivo between circulating immune cells and tissue-resident cells, including those of the productively infected respiratory tract, we hypothesize that it approaches the situation in mildly infected individuals with a transient, rapidly controlled phase of virus replication involving a limited amount of virus-induced cell damage and immune dysregulation.” (Lines 387-397)

- If the authors would want to expand their observations of the interactions between PBMCs, infectious virions and signaling (mainly cytokines) from productively infected epithelial cells, they could try to infect Calu-3 cells with SARS-CoV-2, collect the supernatant after 48 hours and expose fresh PBMCs to it. They could do this in two conditions, one with supernatant containing cytokines and infectious virus, and the other one containing only cytokines after treatment with a detergent at low concentration (ex: Triton) that would kill the virus but not the cytokines.

Reply: Thank you for the excellent suggestion that is shared with Reviewers #1 and #3. We performed a series of additional experiments addressing this comment, with the corresponding data summarized in new Figure EV5. To mimic the environment of circulating PBMCs of a SARS-CoV-2-infected individual, we individually pre-treated PBMCs with type I IFN, with supernatant from SARS-CoV-2-infected lung epithelial cell cultures, and with serum

obtained at an early stage post-infection from mildly COVID-19 diseased individuals. The two latter treatments, but not type I IFN treatment, statistically significantly enhanced the cell-associated quantities of viral RNA upon exposure, suggesting cytokines induced by SARS-CoV-2 infection *in vitro* and *in vivo* may favor interaction of PBMCs with virus particles. However, the two treatments did not modulate cellular gene expression to detectable levels, while IFN, as expected, upregulated *IFIT1* mRNA expression. Together, these new results suggest that PBMCs that are “primed” by stimulation with a cytokine milieu that is characteristic of an ongoing systemic SARS-CoV-2 infection display a slightly increased ability to physically interact with viral particles, in the absence of detectable changes of *IFIT1* and *IL6* mRNA expression:

*“Finally, we were intrigued whether pre-activation would result in an altered ability of PBMCs to interact with, internalize and sense SARS-CoV-2, as opposed to freshly isolated PBMCs from healthy donors. To mimic the environment of circulating PBMCs of a SARS-CoV-2-infected individual, we individually pre-treated PBMCs with type I IFN, with supernatant from SARS-CoV-2-infected lung epithelial cell cultures, and with serum obtained at an early stage post-infection from mildly COVID-19 diseased individuals (Fig. EV5). IFN- α 2a pre-treatment did not influence the amount of detectable viral RNA, neither in the cell-associated fraction (Fig. EV5A, left panel) nor in the supernatant (Fig. EV5B, left panel), despite expected IFN-mediated enhancement of *IFIT1* (Fig. EV5C) but not *IL6* mRNA (Fig. EV5D) expression.*

*Pre-treatment of PBMCs with virus-free, cytokine-containing supernatant derived from SARS-CoV-2-infected Calu-3 cell cultures resulted in a mild (1.6-fold), but statistically significant increase of cell-associated viral RNA copies (Fig. EV5B, middle panel) when compared to cells stimulated with supernatant from uninfected Calu-3 cells, in the absence of changes concerning viral RNA quantities (Fig. EV5A, middle panel) in the culture supernatant, as well as *IFIT1* and *IL6* mRNA expression (Fig. EV5C-D, middle panels).*

*Similarly, cultivation of PBMCs in the presence of serum from COVID-19 patients, as opposed to mock or control serum treatment prior to SARS-CoV-2 exposure, was followed by an 1.8-fold higher abundance of viral genomic RNA in the cellular fraction (Fig. EV5B, right panel). In summary, these results suggest that PBMCs that are “primed” by stimulation with a cytokine milieu that is characteristic of an ongoing systemic SARS-CoV-2 infection display a slightly increased ability to physically interact with viral particles, in the absence of detectable changes of *IFIT1* and *IL6* mRNA expression.” (Lines 289-310)*

“Interestingly, pre-stimulation of PBMCs in a cytokine-containing milieu using virus-free supernatants from infected lung epithelial cell cultures and sera from COVID-19 patients sensitized cells for a slightly more efficient uptake of particles.” (Lines 337-339)

Minor points

- Title: Innate immune responses instead of innate immunity responses?

Reply: We have changed the title accordingly.

- Line 62: define manifestation index

Reply: Manifestation index is the probability of development of a clinically apparent illness upon infection with a given pathogen. By mentioning this term, we initially meant to refer to the complication of managing a pandemic when a significant fraction of infected persons get infected in the absence of symptoms, while nevertheless shedding the virus. We have now deleted this part of the sentence because this issue is included in the phrase “*pre- and asymptomatic infectious phases*” (Lines 76-77) of the identical sentence.

- Line 72: the authors may want to add paloxavir now

Reply: we included Paxlovid: “the protease inhibitor Paxlovid (Hammond et al. 2022)” (Lines 85-86).

- Fig. 1: In the figure, the authors sometimes use the word "mock" to refer to uninfected, untreated cells and sometimes to refer to untreated, SARS-CoV-2 infected cells. This is confusing at times.

Reply: Correct. We clarified this issue where appropriate by changing to “mock-exposed” in all figures and by clarifying in the text.

- Fig. 1A and 1B: why do the authors not show the data for timepoints earlier than 72 hpe?

Reply: we now show data of timepoints 8h, 12h, 24h, 48h, corroborating the absence of evidence for *de novo*-generated infectivity.

- Line 117 - 118: In relation to previous comment, the authors talk about infectivity at 24 hours but this datapoint is not shown?

Reply: Correct, we apologize for this mistake. We corrected this phrase as follows: “Treatment of cells with the polymerase inhibitor Remdesivir did not further reduce infectivity in the supernatant, suggesting that the infectivity detectable in the mock-treated, virus-exposed cultures reflects virus input (**Fig. 1B**).” (Lines 128-130)

- the authors may want to address why in Fig. 1A there is no virus at 72 hpe but there is plenty of virus at 72 hpe in Fig. 1B?

Reply: Thanks for pointing out this apparent inconsistency. We believe that the differences observed in the decay of virus infectivity (PFU/ml) in the supernatant originate from the slightly different inocula, 33500 PFU/ml and 140750 PFU/ml for Figure 1A and B, respectively. However, both experiments support our main hypothesis that the infectivity of the cell culture supernatant constantly decreases without evidence for virus replication.

- Line 256: remove "in line with"

Reply: “in line with” is deleted.

- Line 260-263: homogenize font size

Reply: Font is fixed.

- Line 430: remove "extraction"

Reply: "extraction" is removed.

- Line 456 - 458: homogenize font

Reply: Font is fixed.

- The reference for Corman et al., 2020 is missing

Reply: Citation is added.

Reviewer #3:

This manuscript describes an *ex vivo* study of SARS-CoV and SARS-CoV-2 infection in PBMCs, which is important to the understanding of the host cell spectrum of Sarbecoviruses and evaluating the roles of virus+ myeloid cells observed in clinical samples. The manuscript is well written. I have three concerns about the data analysis.

1. The authors should analyze the detectability of different ORFs in PBMCs. SARS-CoV-2 RNAs can be detected in myeloid cells derived from clinical samples and demonstrated a gradient from the 5' end to 3' end along the viral genome, indicating subgenomic transcription. I am curious whether *ex vivo* studies could reproduce this observation.

Reply: Thank you for this suggestion. In addition to sgE, we now show subgenomic N mRNA levels (Fig. EV1B).

2. One conclusion of this study is that physical interaction and internalization of SARS-CoV-2 with monocytes can induce interferon responses. I am curious what components of the viral particles are important to induce the interferon responses of monocytes and whether internalization is necessary. The authors are expected to discuss whether cytokine stimulations could sensitize the internalization of myeloid cells so that SARS-CoV-2 RNA could be detected more frequently in macrophages and neutrophils.

Reply: Thank you for the excellent suggestion that is shared with Reviewers #1 and #2. We performed a series of additional experiments addressing this comment, with the corresponding data summarized in new Figure EV5. To mimic the environment of circulating PBMCs of a SARS-CoV-2-infected individual, we individually pre-treated PBMCs with type I IFN, with supernatant from SARS-CoV-2-infected lung epithelial cell cultures, and with serum obtained at an early stage post-infection from mildly COVID-19 diseased individuals. The two latter treatments, but not type I IFN treatment, statistically significantly enhanced the cell-associated quantities of viral RNA upon exposure, suggesting cytokines induced by SARS-CoV-2 infection *in vitro* and *in vivo* may favor interaction of PBMCs with virus particles. However, the two treatments did not modulate cellular gene expression to detectable levels, while IFN, as expected, upregulated *IFIT1* mRNA expression. Together, these new results suggest that PBMCs that are "primed" by stimulation with a cytokine milieu that is characteristic of an ongoing systemic SARS-CoV-2 infection display a slightly increased ability to physically interact with viral particles, in the absence of detectable changes of *IFIT1* and *IL6* mRNA expression:

“Finally, we were intrigued whether pre-activation would result in an altered ability of PBMCs to interact with, internalize and sense SARS-CoV-2, as opposed to freshly isolated PBMCs from healthy donors. To mimic the environment of circulating PBMCs of a SARS-CoV-2-infected individual, we individually pre-treated PBMCs with type I IFN, with supernatant from SARS-CoV-2-infected lung epithelial cell cultures, and with serum obtained at an early stage post-infection from mildly COVID-19 diseased individuals (Fig. EV5). IFN- α 2a pre-treatment did not influence the amount of detectable viral RNA, neither in the cell-associated fraction (Fig. EV5A, left panel) nor in the supernatant (Fig. EV5B, left panel), despite expected IFN-mediated enhancement of IFIT1 (Fig. EV5C) but not IL6 mRNA (Fig. EV5D) expression.

Pre-treatment of PBMCs with virus-free, cytokine-containing supernatant derived from SARS-CoV-2-infected Calu-3 cell cultures resulted in a mild (1.6-fold), but statistically significant increase of cell-associated viral RNA copies (Fig. EV5B, middle panel) when compared to cells stimulated with supernatant from uninfected Calu-3 cells, in the absence of changes concerning viral RNA quantities (Fig. EV5A, middle panel) in the culture supernatant, as well as IFIT1 and IL6 mRNA expression (Fig. EV5C-D, middle panels).

Similarly, cultivation of PBMCs in the presence of serum from COVID-19 patients, as opposed to mock or control serum treatment prior to SARS-CoV-2 exposure, was followed by an 1.8-fold higher abundance of viral genomic RNA in the cellular fraction (Fig. EV5B, right panel). In summary, these results suggest that PBMCs that are “primed” by stimulation with a cytokine milieu that is characteristic of an ongoing systemic SARS-CoV-2 infection display a slightly increased ability to physically interact with viral particles, in the absence of detectable changes of IFIT1 and IL6 mRNA expression.” (Lines 289-310)

“Interestingly, pre-stimulation of PBMCs in a cytokine-containing milieu using virus-free supernatants from infected lung epithelial cell cultures and sera from COVID-19 patients sensitized cells for a slightly more efficient uptake of particles.” (Lines 337-339)

3. The current trajectory analysis is weird. SARS-CoV treated samples should not be mixed with SARS-CoV-2 treated samples to infer potential developmental trajectories.

Reply: We acknowledge that trajectory analysis is more commonly used to track cell development and differentiation. However, in our case, we wanted to use this tool to understand how PBMCs exposed to these different viruses develop relative to each other. That is, in this analysis, the goal was not merely to understand how cells exposed to SARS-CoV or SARS-CoV-2 develop respectively as a result of the exposure, but rather whether cells that have been exposed to either virus follow the same or a different trajectory as a result of viral exposure. Indeed, our analysis indicates that, for the most part, SARS-CoV-2 exposed cells branch off and follow a different trajectory, defined largely by ISG expression, compared to SARS-CoV exposed cells which follow a trajectory more similar to that of the mock exposed cells. The only way to show clearly that differential exposures lead to a branched cell fate and not, say, to a development along the same track, is to “mix” the cells from different treatments and infer their fates relative to each other. We have clarified our approach by adding this explanation in the results’ section:

“For each cell type, cells from all three treatments were subclustered and genes differentially expressed between clusters were used as input for cell trajectory analysis using the Pseudotime algorithm from the monocle R package (Trapnell et al. 2014). We aimed to identify whether cells from different treatments, especially those exposed to different viruses, developed along the same trajectory as a result of the exposure or if a different cell fate was induced (Fig. 4A).” (Lines 219-224)

Overall, this study provides an ex vivo study of SARS-CoV and SARS-CoV-2 infection in PBMCs, which is insightful and important to understand the biology of SARS-CoV-2.

27th Jul 2022

Manuscript Number: MSB-2022-10961R

Title: Non-productive exposure of PBMCs to SARS-CoV-2 induces cell-intrinsic innate immune responses

Author: Julia Kazmierski

Kirstin Friedmann

Dylan Postmus

Jackson Emanuel

Cornelius Fischer

Jenny Jansen

Anja Richter

Laure Bosquillon de Jarcy

Christiane Schüler

Madlen Sohn

Sascha Sauer

Christian Drosten

Antoine-Emmanuel Saliba

Leif Sander

Marcel Müller

Daniela Niemeyer

Christine Goffinet

Dear Christine,

Thank you for sending us your revised manuscript. We have now received the comments from the three referees who were asked to re-review your manuscript. As you will see below, the referees are satisfied with the modifications and think the study is now suitable for publication.

Before we can formally accept your manuscript, we would ask you to address the following editorial-level issues:

When you resubmit your manuscript, please download our CHECKLIST (<https://bit.ly/EMBOPressAuthorChecklist>) and include the completed form in your submission. *Please note* that the Author Checklist will be published alongside the paper as part of the transparent process (<https://www.embopress.org/page/journal/17444292/authorguide#transparentprocess>)

Click on the link below to submit your revised paper.

Thank you for submitting this interesting paper to Molecular Systems Biology.

Kind regards,

Jingyi

Jingyi Hou

Editor

Molecular Systems Biology

Reviewer #1:

Reviewers comments were addressed. No further points of critique.

Reviewer #2:

The authors have successfully addressed all my comments and I believe the quality of the manuscript has improved. I find the work suitable for publication.

Reviewer #3:

The authors have addressed all my concerns.

The authors performed the requested changes.

28th Jul 2022

Manuscript number: MSB-2022-10961RR

Title: Non-productive exposure of PBMCs to SARS-CoV-2 induces cell-intrinsic innate immune responses

Dear Dr. Goffinet,

Thank you again for sending us your revised manuscript. We are now satisfied with the modifications made and I am pleased to inform you that your paper has been accepted for publication.

*** PLEASE NOTE *** As part of the EMBO Publications transparent editorial process initiative (see our Editorial at <https://dx.doi.org/10.1038/msb.2010.72>), Molecular Systems Biology publishes online a Review Process File with each accepted manuscripts. This file will be published in conjunction with your paper and will include the anonymous referee reports, your point- by-point response and all pertinent correspondence relating to the manuscript. If you do NOT want this File to be published, please inform the editorial office at msb@embo.org within 14 days upon receipt of the present letter.

Should you be planning a Press Release on your article, please get in contact with msb@wiley.com as early as possible, in order to coordinate publication and release dates.

LICENSE AND PAYMENT:

All articles published in Molecular Systems Biology are fully open access: immediately and freely available to read, download and share.

Molecular Systems Biology charges an article processing charge (APC) to cover the publication costs. You, as the corresponding author for this manuscript, should have already received a quote with the article processing fee separately. Please let us know in case this quote has not been received.

Once your article is at Wiley for editorial production you will receive an email from Wiley's Author Services system, which will ask you to log in and will present you with the publication license form for completion. Within the same system the publication fee can be paid by credit card, an invoice or pro forma can be requested.

Payment of the publication charge and the signed Open Access Agreement form must be received before the article can be published online.

Molecular Systems Biology articles are published under the Creative Commons licence CC BY, which facilitates the sharing of scientific information by reducing legal barriers, while mandating attribution of the source in accordance to standard scholarly practice.

Proofs will be forwarded to you within the next 2-3 weeks.

Thank you very much for submitting your work to Molecular Systems Biology.

Sincerely,
Jingyi

Jingyi Hou, PhD
Scientific Editor
Molecular Systems Biology